# Occupational Exposures and Environmental Health Hazards of Military Personnel

**DOI:** 10.3390/ijerph18105395

**Published:** 2021-05-18

**Authors:** Marta Geretto, Marco Ferrari, Roberta De Angelis, Filippo Crociata, Nicola Sebastiani, Alessandra Pulliero, William Au, Alberto Izzotti

**Affiliations:** 1Department of Experimental Medicine, University of Genoa, 16132 Genoa, Italy; marta_geretto@hotmail.it; 2Texas Biomedical Research Institute, 8715 W. Military Drive, San Antonio, TX 78227, USA; mferrari@txbiomed.org; 3Department of Oncology and Molecular Medicine, Istituto Superiore di Sanità, 00161 Rome, Italy; roberta.deangelis@iss.it; 4General Inspectorate of Military Health, 00184 Rome, Italy; consulente@igesan.difesa.it (F.C.); ispettore@igesan.difesa.it (N.S.); 5Department of Health Sciences, University of Genoa,16132 Genoa, Italy; alessandra.pulliero@unige.it; 6Faculty of Medicine, Pharmacy, Science and Technology University of Medicine, Pharmacy, Science and Technology, 540142 Targu Mures, Romania; wau@stu.edu.cn; 7Department of Preventive Medicine and Community Health, University of Texas Medical Branch, Galveston, TX 77555, USA; 8IRCCS Ospedale Policlinico San Martino, 16132 Genoa, Italy

**Keywords:** molecular epidemiology, biomarkers, soldiers, environmental pollution, occupational exposures

## Abstract

Background: Military personnel are frequently exposed to environmental pollutants that can cause a variety of diseases. Methods: This review analyzed publications regarding epidemiological and biomonitoring studies on occupationally-exposed military personnel. Results: The exposures include sulfur mustard, organ chlorines, combustion products, fuel vapors, and ionizing and exciting radiations. Important factors to be considered are the lengths and intensities of exposures, its proximity to the sources of environmental pollutants, as well as confounding factors (cigarette smoke, diet, photo-type, healthy warrior effect, etc.). Assessment of environmental and individual exposures to pollutants is crucial, although often omitted, because soldiers have often been evaluated based on reported health problems rather than on excessive exposure to pollutants. Biomarkers of exposures and effects are tools to explore relationships between exposures and diseases in military personnel. Another observation from this review is a major problem from the lack of suitable control groups. Conclusions: This review indicates that only studies which analyzed epidemiological and molecular biomarkers in both exposed and control groups would provide evidence-based conclusions on exposure and disease risk in military personnel.

## 1. Introduction

Military personnel are exposed to environmental hazardous substances that can be very different from those for industry workers and for the general population (Figure 1). However, individual response to occupational hazards is extremely variable due to complex interplay among several factors, such as individual health, genetics, and time of exposure. Environmental samplings can be used to provide information regarding the presence of hazardous substances, while it is ineffective to determine individual exposure and response [1]. To investigate relationships occurring between military health and warfare theater environment, it is necessary to use an integrated approach based on both analytical and molecular epidemiology. To assess whether a specific agent impacts on human health, it is necessary to make a complex evaluation that includes exposure, association, and effect indexes. For this purpose, it is important to adopt integrated monitoring programs which are able to integrate information resulting from analyses of exposure biomarkers, biomarkers of effect and biomarkers of susceptibility. Exposure biomarkers are represented by chemicals or their metabolites found in biological samples (urine, blood, breath, and saliva) [2]. The main utilities of this class of biomarkers are related to their ability to prove that exposure has occurred, to identify the route of exposure, and the absorbed dose of chemicals considered. Body burdens from exposure to many of the substances in military personnel are often difficult to be realized due to the remarkable logistic problems existing in the operative theaters where the priority is the military activity. Biomarkers of early effect can provide information dealing with the biological consequences from an exposure and with time intervals, as typically occurs for carcinogens [3]. Biomarkers of effect are metabolites or molecules which are capable of inducing development of diseases, while biomarkers of susceptibility are factors, such as genetic polymorphisms which are able to modify individual’s susceptibility to chemical exposures. The main characteristics of a good biomarker are sensitivity, specificity, and noninvasive sampling collection methods. Nevertheless, there are some sources of error. Indeed, some chemicals are characterized by short half-lives in the human body which influence their detection and sample timings.

In addition, it is necessary to perform pre- and post-deployment samplings to establish a cause–effect relationship between specific exposures and symptoms or disease onset. Indeed, there may be underlying diseases prior to pollutant exposures. In particular, this has been emphasized by Skabelund and Morris [4,5], who performed pre- and post-deployment spirometry studies on military personnel who were dispatched to southwest Asia, where exposure to geologic dusts and burn pit emissions could be related to lung injuries.

Another question with population monitoring is represented by the high level of interpersonal and intrapersonal variabilities that would require increasing the size of both exposed and control groups [6]. Cross-sectional studies comparing data in two different groups are greatly affected by similar confounding factors. The study design most useful to attenuating these confounding factors is prospective cohort studies which examine the same endpoints before and after theater deployments in the same military population.

Military exposures are of interest for public health purposes. Indeed, military personnels may be exposed to high-risk substances at high doses, thus providing reliable toxicological information of great relevance for managing environmental-related risk in the general population. Particular problems arise when military personnel are deployed in dense urban operational environments and their outskirts. Indeed, under these circumstances, military personnel face occupational hazards due to overcrowding, such as toxic industrial chemicals, water pollution, traffic- and industrial-related air pollution, etc. [7].

Over the last few years, several studies have been performed on military forces around the world, but only a few of them analyzed all the three indexes regarding exposure, association, and effect, as previously mentioned. Moreover, biomarker analyses assume a priori selection of chemicals to be measured. This hampers the detection of other molecules or metabolites that could be present in each sample, thus missing any information of other exposures. In addition, even if biomonitoring is an essential tool for military personnel surveys, most of the chemicals are still unknown under the toxicological profile. For these reasons, the omic technologies (genomics, metabolomics, proteomics, lipidomics, and transcriptomics) which can be applied to environmental health research are useful to characterize both external and internal exposure to different pollutants [8]. However, some methodological approaches can be specifically adopted to overcome this problem by evaluating the overall impact of environmental pollutions as a whole instead of as single pollutants. An example of this approach is the analysis of DNA adductome by ^32^P post-labeling which is capable of detection with high sensitivity (1 adduct every 10 cells), as induced by exposure to mixtures also including unknown genotoxic pollutants [9].

The aim of this review was to analyze different toxic compounds affecting soldiers in warfare zones and operational theaters, with a focus on biomarkers in each study. For some studies, long-term follow-up of exposed soldiers has been completed, thus producing valuable epidemiologic data on health indicators in military personnel.

## 2. Methods

Articles were selected through a literature search in the electronic database PubMed using the following research strategy: ((“Military Personnel”[Mesh]) AND “Environmental Monitoring”[Mesh]) AND “Environmental Pollutants”[Mesh]) AND “Environmental Exposure”[Mesh])).

Abstracts were analyzed and selected for further analysis when studies were performed on veterans or active soldiers. Finally, investigators included additional relevant papers after reference cross-check.

### 2.1. Literature Search

In order to identify all potentially eligible studies, a comprehensive search on the National Library of Medicine’s PubMed online catalogue and the Cochrane Database of Systematic Reviews from inception up to 1964 were conducted, and an update was carried out in 2020.

A search strategy was developed and adapted for each database, using a combination of the following keywords: military personnel exposed, biomonitoring, genotoxic pollutants, soldiers, occupational exposures, and molecular epidemiology, applied as Mesh terms. Unpublished and grey literature data were not sought.

### 2.2. Inclusion and Exclusion Criteria

Studies were considered eligible in the systematic review if they met the following criteria: provided information on military personnel exposed to genotoxic pollutants. Only original articles published in English and in peer-reviewed journals were eligible for data extraction. Animal and in vitro studies were not included. Letters, comments, editorials, and case reports were also excluded.

### 2.3. Identification and Selection of Studies

From the collected publications, primary titles and abstracts were initially scanned by one of our authors who identified publications for request of full-text articles for further review. When the full text of an article was obtained, it will be reviewed by two of our authors. Subsequently, inclusion or exclusion of each study was determined by discussion and a consensus between the two. Any disagreements were resolved through discussions or after consultation with a third author. Article bibliographies were also inspected in order to identify any additional studies that might have been missed during the initial search. Sponsors were not contacted. The literature retrieval and selection procedures were in adherence to the Preferred Reporting Items for Systematic Reviews and Meta-Analyses (PRISMA) statement (Figure 2).

## 3. Sulfur Mustard

Mustard gas or sulfur mustard (SM) was synthesized for the first time in 1822, but it has been used as a chemical warfare agent only since the battle of Ypres in 1917, during World War I. SM, bis (2-chloroethyl) sulfide, is an alkylating agent with well-characterized acute effects. It is known to be a strong blister agent, affecting mainly the respiratory tract by inducing acute inflammation, necrosis of the mucosa, and tracheal stenosis. It also affects cutaneous tissues, where the commonly observed symptoms in SM-exposed subjects are erythema, hypo- and hyper-pigmentation, blisters, ulcers, and edema. Furthermore, SM causes severe eye complications, including conjunctivitis and corneal erosion and opacity. Other acute effects related to mustard gas exposure include reproductive and sleep disorders [10,11,12,13]. Dealing with SM long-term effects, IARC has classified SM in class 1 human carcinogens [14].

During the Iran–Iraq war in 1980–1988, the Iraqi army used SM against Iranian troops and civilians, causing more than a hundred thousand casualties with acute and sub-acute toxic effects [15,16]. Since then, Iranian veterans have been followed-up to verify long-term effects of SM-exposure in combatants. A study performed on 236 soldiers after 2–28 months from exposure showed complications affecting the respiratory tract in 78% of the cases, the central nervous system in 45%, the skin in 41%, eyes in 36%, and other effects at lower frequencies [17]. Other studies explored the long-term effects (after 15–20 years) of SM-exposure on Iranian veterans which demonstrated that the main complications were in lungs (42.5–100%), peripheral nerves (77–77.5%), eyes (39.3–77.61%), and skin (24.5–82.84%) [18,19,20,21]. High frequency variations found in these studies were due to differences in sample size (43 to 34,000 subjects analyzed), but nevertheless, the major pathological events were the same. Since respiratory complications were the main cause of SM-related disability among exposed veterans [22], several studies were focused on the respiratory apparatus. The most frequent long-term respiratory complications detected in the upper tract were dysphonia (79.1% of subjects), post-nasal discharge (41.9%), lower larynx position (30.2%), limitation of vocal cords (25.6%), and inflammation of larynx mucosa (14.8%), while in the lower tract, chronic obstructive pulmonary disease (COPD) (35–84%), bronchiectasis (32.5–44.1%), asthma (25%), large-airway narrowing (15%), pulmonary fibrosis (7.5–7.7%), and simple chronic bronchitis (5%) were the most frequent [23,24]. After about 25 years from the exposure, bronchiectasis (25%), pulmonary fibrosis (25%), and ground-glass attenuation (16.66%) were diagnosed in 43 Iranian veterans, causing pulmonary function test abnormalities in 44.18% of the subjects investigated [22]. In the descriptive study, 128 SM-exposed veterans with severe eye injury were compared with 31 healthy controls. Serum levels of IL-1α and Fas Ligand (FasL), a homotrimeric type II transmembrane protein expressed on cytotoxic T lymphocytes, were significantly higher among the cases than among the controls (*p* < 0.001 and *p* = 0.037, respectively). Additionally, a significant decrease was observed in serum and tear levels of TNF-α in the cases as compared with controls (*p* < 0.001 and *p* < 0.001, respectively). Serum levels of FasL were significantly higher in cases with severe ocular involvement than in the controls (*p* = 0.03). Serum levels of IL-1α and FasL were reported to cause different ocular surface abnormalities in sulfur mustard-exposed patients [25]. The SM-exposure could alter immunoglobulins level compared with healthy controls and the changes of IgG2 and IgG1 levels were associated with some ocular problems [26].

Moreover, an increased risk of infections and tumors which were observed in these subjects were associated with an impaired immune system, with a significant difference in the percentage of monocytes, CD3^+^ T-lymphocytes, and CD16^+^56^+^ cells, difference in IgM and C3 levels, and beta_2_ and gamma globulins in 40 Iranian veterans’ serum compared to an unexposed control group [27,28].

Several studies in the last few years were focused on the identification of SM-exposure biomarkers, and on cellular and molecular mechanisms involved in SM-related pulmonary pathologies. Expression analyses of genes involved in oxidative stress and antioxidant defense in biopsies from 6 lungs of SM-exposed subjects (after 25 years from the event) revealed a disrupted expression pattern for more than eighty genes. In particular, the most upregulated genes were peroxiredoxins (PRDXS) and sulfiredoxin-1 (SRXN1) [29], oxidative stress responsive kinase-1 (OXSR1), forkhead box M1 (FOXM1), glutathione peroxidase-2 (GPX2) [30], and reactive oxygen species (ROS); in association with aldehyde oxidase 1 (AOX1), myeloperoxidase (MPO), dual oxidase 1 and 2 (DUOX1, DUOX2), thyroid peroxidase (TPO), and eosinophil peroxidase (EPO) [31]. On the other hand, the most downregulated genes were metallothionein-3 (MT3) and glutathione reductase (GSR). GSR downregulation was associated with reduced activity of GSH-dependent antioxidant enzymes such as glutathione transferases (GSTs), glutathione peroxidases (GPXs), and sulfiredoxin-1 (Srx1) [13,29]. These alterations clearly indicate that oxidative stress had a major role in long-term pulmonary pathologies decades after exposure. Glutathione-S-transferase (GST) activity and vitamin C were significantly decreased in sulfur mustard-exposed patients as compared with controls. Besides, Cu level and Cu/Zn ratio in sulfur mustard-exposed veterans showed a significant correlation with the severity of the diseases [32]. The recruitment of leukocytes at the site of SM-injury caused the production of EPO and MPO enzymes, with ROS accumulation and consequent oxidative damage to DNA, lipids, and proteins of lung cells. Other studies investigated lipid peroxidation derivative malondialdehyde (MDA) levels as an oxidative stress measure in serum, and 8-oxo-dG genomic DNA content and OGG1 expression as biomarkers for oxidative damage in 215 veterans, at 25 years after exposure [13,33]. Increased MDA levels indicated oxidative stress in poisoned subjects, confirming the results of a historical cohort investigation by Behravan et al. [34] on 40 veterans who showed increased serum levels of 8-isoprostane F2-alpha. Behboudi and colleagues [33] demonstrated that 8-oxo-dG and OGG1 mRNA expression levels were increased, when compared to a control group, indicating a higher oxidative damage in SM-exposed veterans. Additionally, the length of telomeres in leukocytes and p16^INK4a^ mRNA expression were investigated as biomarkers for cellular senescence. Length of telomeres in leukocytes was shown to be significantly shorter in exposed veterans than in non-exposed controls, in line with data reported by Behravan et al. [34]. The expression level of p16^INK4a^ was lower in exposed compared to non-exposed subjects, indicating an impaired immune system and cellular senescence [33]. DNA damage was confirmed by another cross-sectional study performed by Katheri and colleagues [18] on 40 SM-exposed Iranian veterans, showing the same long-term complications previously described (respiratory, ocular, and cutaneous pathologies). The higher levels of phosphor-H2AX, a histonic DNA damage biomarker, were not significant, but were consistent with the results of Behboudi and colleagues. Furthermore, the four DNA repair proteins (MRE11, NBS1, RAD51, and XPA) showed lower expressions in SM-exposed subjects, confirming the persistence of DNA damage and impaired repair mechanism 25 years after the intoxication episode [18,33].

MicroRNA expressions were evaluated in a case-control study on 84 veterans who were split into 4 groups according to COPD severity. The results showed that several microRNAs were characterized by an altered pattern of expression. Most of the microRNAs were downregulated, but some of them showed upregulation in the mild and severe cases: miR-589-3p, miR-365a-3p, miR-143-3p, miR-200a-3p, and miR-663a. These microRNAs are normally altered in hypoxic and oxidative stress conditions or are involved in cell cycle and chronic inflammation regulation. The authors highlighted that the TGF-beta signaling pathway, cell cycle arrest, and apoptosis and senescence were the major mechanisms triggered in SM-exposed veterans, and proposed miR-143-3p as a candidate biomarker for SM in order to discriminate SM-induced pathologies from similar pathologies caused by other risk factors [35]. An overview of the main studies dealing with the sulfur mustard-exposures of military personnel is reported in Table 1.

The previously described studies did not quantify the toxic agent exposures. For this reason, the authors just split the subjects into groups based on the observed severity of the SM-related pathologies. Accordingly, the main limitations of these studies are the lack of evaluation of the environmental and individual exposure doses to SM. Furthermore, all the studies were conducted on male soldiers, resulting in a lack of information about the sanitary and molecular effects of SM on female combatants.

## 4. Sarin and Cyclosarin

Immediately after the end of the Gulf War in 1991, destruction of an ammunition storage in Kashiwa, Iraq potentially exposed hundreds of thousands of US soldiers to Sarin (GB) and Cyclosarin (GF). These nerve agents can be absorbed through inhalation or by mucosa, skin, or eyes. At very high concentrations, they can be lethal. Studies performed in 1999 and 2001 [37,38], based on a 1997 plume dispersion model, did not detect an increased risk of mortality or unusual morbidity in GB/GF-exposed soldiers after several years. Bullman and colleagues [39] performed a 10-year follow-up, from 1991 to 2000. They determined the exposure likelihood of 351,121 soldiers and identified 100,487 exposed individuals based on a new plume model as described in 2000 by the US Defense Department. Their analyses confirmed the previous results but highlighted a statistically significant increased risk in brain cancer deaths among exposed veterans (RR = 1.94). Relative risk reached 3.26 in soldiers exposed for 2 days, while it was 1.72 for 1-day-exposed individuals [39]. GB and GF are important agents affecting brain structure, reducing white and grey matter density, increasing axial diffusivity in parietal cortex, capsules, front-occipital fascicule and corona radiata, and increasing hippocampal volume [40,41,42,43,44]. An association between high levels of GB/GF-exposure, white matter reduction, and ventricles volume increase has been demonstrated by Heaton and colleagues [44] in a study involving 13 exposed veterans, compared to 13 non-exposed controls. This result could explain the dose–response relationship in neurobehavioral performances observed in 140 veterans [45]. Decreased performance and delayed response time in neuro-psychological tests have been confirmed in subsequent studies [40,41]. Chao and Zhang suggested that hippocampus alterations in Gulf War veterans could be related to premature aging, and these modifications are associated with an increased risk of late-life dementia [43]. Soldiers were given pyridostigmine bromide (PB) as a pre-treatment to protect from nerve gas agents [46]. PB exposure blocked the action of acetylcholinesterase and potentiates cholinergic activation, similar to the action of the organophosphate and carbamate group of pesticides [46]. Pyridostigmine is the current pretreatment for nerve agent poisoning and is in use by most of the armed forces in Western countries. However, since pyridostigmine barely crosses the blood–brain barrier, it provides no protection against nerve agent-induced central injury [45]. Pyridostigmine is ineffective when administered without post-exposure treatment adjuncts. Therefore, other directions for prophylactic treatment should be explored. These include combinations of carbamates (reversible AChE inhibitors) and central anticholinergics or NMDA receptor antagonists, benzodiazepines or partial agonists for benzodiazepine receptor, and other central AChE inhibitors approved for Alzheimer’s disease. The main problem of these studies is that no early or late exposure biomarkers were analyzed in veterans, in comparison to investigations on the victims of the 1995 Tokyo subway attack, where the levels of serum cholinesterase were decreased after more than 5 years from the exposure [46,47]. Furthermore, most of the papers had small sample sizes (13–140 exposed subjects) compared to the total of the soldiers potentially exposed (Table 2).

## 5. Organochlorinates

### 5.1. Herbicides

During the Vietnam War, the US army extensively applied herbicides using UC-123 aircrafts to defoliate forests and US-base perimeter foliage to increase security. Herbicides were also used to eradicate unwanted vegetation inside the military installations, resulting in potential prolonged exposures for American soldiers. Aircraft contamination was also an important route of dioxin-contaminated Agent Orange (AO/TCDD) exposure for flight crews and maintainers even after years from the end of the war, as demonstrated by Lurker et al. [48], who evaluated dermal and respiratory intake, and dioxin contamination levels inside vehicles. They estimated a body-burden for contact exposure of 0.92 and 5.4 pg/kg body-weight-day for the crews and for maintainers, respectively. Inner aircraft concentrations ranged from 11.49 to 13.2–27.0 pg/m^3^, depending on the theoretical model applied.

On the other side, Ross and colleagues [49] highlighted that exposures in military ground installations were probably less important than expected because of several factors. As an example, intervals between AO reapplications were longer than they should have been, with more than a one-year interval in bigger bases. Furthermore, facilities were usually far enough from the sprayed perimeter. Agent Orange presented TCDD manufacturing contaminations, with concentration varying between 0.02 and 47 ppm (average 2 ppm in 1965–1970 timeframe), depending on the manufacturer and the production date [49]. Ross and colleagues estimated the dosage of TCDD for AO applicators and bystanders. The dosages for applicators resulted higher than for other combatants, but the burden of TCDD fell in the range of civilian applicators in the US in the same period.

Epidemiological studies were performed on Korean and US veterans deployed in Vietnam and exposed to AO/TCDD. Cypel and Kang [50] highlighted a slight non-statistically significant increased risk for all causes of death (adjusted relative risk, ARR = 1.13), all cancers combined (ARR = 1.15), diseases affecting the respiratory system (ARR = 2.20), and an important and statistically significant increased risk for COPD (ARR = 4.82), comparing 2872 exposed to 2737 non-exposed US veterans. Two studies performed on a cohort of more than 180,000 Korean veterans showed an increased mortality for all causes of death in the high-AO-exposed group compared to the less exposed ones (adjusted hazard ratio, aHR = 1.10). Increased mortality was due to stomach (aHR = 1.14–1.17), liver (aHR = 1.12), small intestine (aHR = 2.30–2.88), larynx (aHR = 1.28), lung (aHR = 1.15), thyroid gland (aHR = 11.31), bladder (aHR = 2.04), mouth (aHR = 2.54), and salivary gland cancer (aHR = 6.96). Moreover, an increased risk of death was detected for chronic myeloid leukemia (aHR = 7.91), angina pectoris (aHR = 2.34), COPD (aHR = 1.73), and liver diseases (aHR = 1.19) [51,52]. AO-exposure also increased the prevalence of several diseases in heavily exposed Korean veterans when compared with the less exposed. In particular, endocrine disorders (autoimmune thyroiditis, odds ratio (OR) = 1.93, and pituitary gland disorders, OR = 1.43), amyloidosis (OR = 3.02), COPD (bronchiectasis, OR = 1.16, and bronchitis, OR = 1.05), and liver cirrhosis (OR = 1.08) [52] were reported (Table 3).

All the mentioned studies failed to assess a cause–effect relationship between AO/dioxins and the pathologies described since they did not use direct measures of exposure/effect. A serum dioxin test, which is the gold standard for organochlorine-exposure assessment [53], should have been conducted to confer biological plausibility to the pollutant–health effect relationship described in these epidemiological studies. Unfortunately, this test is expensive and was not performed on US veterans, since the Veterans Affairs Department assumed that all of them had been exposed if they were deployed in the Vietnam War in certain periods [53]. In addition, significant associations were found between levels of polychlorinated dibenzo-p-dioxins (PCDD) congeners other than TCDD and gross motor scores in boys. Perinatal exposure of TCDD and other PCDD congeners affected development of language and gross motor skills respectively, in boys at 2 years of age exposed to dioxins originating from Agent Orange in Vietnam. [54].

### 5.2. Pesticides

Soldiers deployed in war theaters are also exposed to vector-borne diseases such as malaria, dengue fever, and Lyme disease. Due to this reason, it is essential to provide an effective protection against blood-feeding arthropods. Common strategies adopted by military forces are insecticide spraying and the use of impregnate uniforms. Pyrethroids have low mammal toxicity, high stability, and long-lasting activity against insects. These features make pyrethroids particularly useful for disease-vector control. However, health concerns could arise due to prolonged exposure to these compounds. Permethrin, used for impregnated bed-nets and clothes, including military uniforms, has oral, cutaneous, and respiratory absorption. Two biomonitoring studies were conducted in the 2003–2005 period on two cohorts (549 and 195 individuals, respectively) of German soldiers [55,56,57]. These studies evaluated urinary pyrethroid metabolite excretion (cis-Cl_2_CA, trans-Cl_2_CA, 3-PBA) and identified higher internal exposures in soldiers with permethrin-impregnated uniforms compared to soldiers with untreated uniforms, with estimations of a 5–6 ug/kg body weight and day internal dose for oral uptake and 1.25 ug/cm^3^ for skin absorption [56]. Moreover, Afghanistan-deployed units showed an increased internal dose compared to Germany-deployed units, probably because of different uniform wearing time (16 h/day and 10 h/day, respectively). Despite the demonstration of insecticide uptake, the authors identified only minor impairments in sensitive subjects, such as sensory discomfort, paranesthesia, skin irritation, and rash. Interestingly, smokers wearing insecticide-treated uniforms presented higher internal levels of permethrin metabolites compared to non-smokers. The authors explained the result as an increased hand–mouth contact [57].

DeBeer et al. [58] investigated whether veterans deployed in Iraq and Afghanistan after the terrorist attacks of 11 September 2001 were exposed to hazardous environmental pollutants. Moreover, the authors were interested in understanding whether these exposures resulted in chronic multisymptomatic illness that were typical of military personnel deployed in the Gulf War. Among the 224 veterans enrolled in this study, 97.2% reported exposures to one or more different pollutants (nerve gas, depleted uranium, pesticides, smoke, or fumes). Despite the small sample size and the absence of a control group, the data show that pesticide exposure was associated to chronic multisymptomatic illness development [58]. Another study evaluated the role of pesticides in the development of chronic neuropsychological dysfunction in Gulf War veterans. High combined exposures were associated with significantly slower information processing reaction times, attentional errors, worse visual memory functioning, and increased mood complaints [59] (Table 4).

### 5.3. Cs Gas (Tear Gas)

Soldiers are exposed to o-Chlorobenzylidene Malononitrile, a CS riot-control agent, both in action and during mask confidence [60,61,62]. Hout et al. showed that all the 6723 components of a US trainee cohort were exposed to CS concentrations higher than the threshold limits of 0.39 mg/m^3^, with 6589 individuals potentially exposed to more than 2.0 mg/m^3^, the immediately dangerous to life and health limit [62]. The same cohort was used to explore associations between CS exposure and clinical outcomes through an observational prospective study [63]: 161 cases of acute respiratory illnesses (ARIs) were diagnosed, 47 prior to CS exposure and 114 after the training. The risk of developing an ARI was higher after exposure (risk ratio (RR) = 2.44), and CS concentrations determined incidences of ARIs. These results clearly indicate an association between exposure and sanitary outcome, but they lacked the analysis of a suitable inner biomarker of exposure/effect. Buchanan and colleagues [60] analyzed the urinary metabolite 2-Chlorohippuric Acid (CHA) levels corrected for creatinine, before and after CS exposure (0.086–4.9 mg/m^3^), in a cohort in 87 US trainee volunteers. The authors demonstrated that urinary CHA levels were not associated with out-of-mask time and suggested that CHA variations could be due to dermal absorption, rather than oral. High CHA levels were found in all subjects after 2 and 8 h from the exposure, and in most of the trainees after 24 and 30 h, with a strong association between CHA, gas concentrations, and times from exposure.

Studies are still required to investigate biological plausibility of CS exposures and health outcomes described by Hout et al. [62,64]. The study of Buchanan et al. [60] on CHA as an exposure biomarker is a first step to reach this goal, but application of biomarkers to identify biological effects is still missing (Table 5).

## 6. Combustion Products

### 6.1. Oil Combustion

During the Desert Storm operation in the 1991 Gulf War, Iraqi troops set fire to about 750 Kuwaiti oil wells, resulting in massive emission of combustion products, with thousands of troops exposed to smoke. Petruccelli and colleagues [64] reported that, in a cohort of 1599 US soldiers deployed in Kuwait in 1991, most of the self-reported symptoms regarded eyes and respiratory tract, with breath shortness, cough, rashes, and fatigue. These symptoms were associated with oil fires’ proximity. The authors indicated the oil-fire smoke as one of the possible factors for the reported symptoms. However, a prospective cohort study on 125 British soldiers could not demonstrate, with objective respiratory measurements, lung function changes before and after deployment [65]. The symptoms of 1560 US veterans deployed in Kuwait in 1991 after 5 years from exposure were evaluated by Lange et al. [66]. The researchers found an increment in ORs for respiratory diseases and depression with increasing self-reported exposures. In contrast, using a GIS modeled exposure, they could not detect any association between symptoms and exposures, concluding that oil-fire smoke did not cause the observed sanitary outcomes.

Known combustion pollutants for Kuwait crude oil included SO_2_, NO_x_, H_2_S, CO, polycyclic aromatic hydrocarbons (PAH), volatile organic compounds (VOCs), metals, and particulates [67].Etzel and Ashley [68] determined VOC concentrations in whole blood of two exposed groups: group I was composed of US personnel employed by the Army or the Army Corps of Engineers in Kuwait City, while group II was composed of medic and paramedic personnel and firefighters working at the burning wells. None of the subjects wore respiratory protections. The investigated VOCs were benzene, ethylbenzene, m-/p-xylene, o-xylene, styrene, and toluene, and the concentration levels detected in groups I and II were compared to those detected in a control group of civilians in the USA. The results demonstrated that no VOC concentration differences occurred among the group I and the control, while group II showed increased levels of ethylbenzene (10 times higher), and more than double concentrations of benzene, m-/p-xylene, o-xylene, styrene, and toluene. Higher levels of VOCs in blood reflected higher air concentrations near the oil fires than in Kuwait City.

Exposure to polycyclic aromatic hydrocarbon (PAH) was also evaluated in two studies by Poirier and colleagues [69] in a cohort of 61 US soldiers stationed in a base in Germany and then deployed in Kuwait from June to September 1991. Urine and blood samples were taken before, during, and after deployment in Kuwait. Bulky aromatic and PAH-DNA adducts were analyzed in 22 blood samples by ^32^P-post-labeling, while urine samples were examined for 1-OH-PG (1-hydroxypyrene-glucuronide) in 33 individuals. Surprisingly, the levels of DNA adducts and 1-OH-PG were lower during deployment than after return to Germany. Low levels of exposure biomarkers were consistent with environmental data obtained by personal sampling pumps and high-volume samplers operating in the burning period. The pumps carried by soldiers did not detect PAHs, while high-volume samplers detected less than half of the 23 investigated PAHs. On the other side, the levels of some carcinogenic PAHs near the US base in Germany were significantly higher than those detected in Kuwait. The authors stated that these values could be due to the industrial activities in the area, to coal combustion for heating in winter, and to dietary intake of PAHs with flame-cooked meat. The studies could not find an influence of CYP1A1 and glutathione S-transferases (GSTM1 and GSTT1) gene polymorphisms on PAH biomarker levels [69]. 

Biological Surveillance Initiative is a useful monitoring program with the aim to evaluate oil well fire pollutants exposure by analyzing markers of internal dose. For this purpose, 3440 soldiers belonging to 11th Armored Cavalry Regiment, home-based in Germany, were interviewed and monitored during pre-deployment in Germany, during deployment in Kuwait, and post-deployment after their return to Germany. Urine and blood samples were collected to analyze the presence of 1-hydroxypirene, VOC, and metals. Moreover, sister chromatid exchanges and DNA-polycyclic aromatic hydrocarbon adducts were evaluated. The results were, however, difficult to understand. Indeed, results of the biomonitoring program performed on soldiers could not be compared to environmental samplings because environmental data were not available. Moreover, interpretation of DNA adducts and sister chromatid exchanges was problematic because of the presence of different confounding factors affecting exposure. Despite these limits, the study is a useful tool for planning future biomonitoring programs [70] (Table 6).

### 6.2. Cooking Oil Fumes

PAHs-exposure can occur because of food habits outside of warfare theaters. A longitudinal study by Lai et al. [71] was performed on 61 Taiwanese military cooks exposed to cooking oil fumes compared to 37 unexposed soldiers. Urine samples were collected at the beginning and at the end of the weekly shift, and urinary levels of 1-OHP (1-hydroxypyrene) and 8-oxo-dG (8-oxodeoxyguanosine) were analyzed and used as a PAH-exposure biomarker and oxidative DNA damage biomarker, respectively. Additionally, PAH concentrations were determined in kitchens and offices, demonstrating higher levels of pyrene, benzo(a)anthracene, benzo(chi)perylene, and median levels of total PAHs. The results showed increased levels of both the biomarkers analyzed at the end of the weekly shift, indicating an association between oxidative stress and cooking oil fumes-exposure in kitchen personnel. Despite emphasis from the authors that mutagenic and carcinogenic compounds emitted from oil fumes were plausible determinants of lung cancer risk, they did not perform any analysis on the health outcomes on military cooks (Table 6).

### 6.3. Open Pit Burning

Solid waste combustion in open pits potentially represents a source of environmental exposure to combustion products for military personnel deployed in warfare theaters. Burn pits (BPs) are usually located in proximity to military bases. Several studies were performed to assess if an association between burn pits smoke-exposure and health outcomes in US military personnel existed.

Prospective studies on the Millennium Cohort participants revealed no increased risk for rheumatoid arthritis or lupus [72], chronic multisymptomatic illness [73], and respiratory outcomes [74].

Masiol et al. [75,76] analyzed PAH, PCDD, and polychlorinated dibenzofurans (PCDF) concentrations at different sites in the Iraqi Joint Base Balad (JBB) in 2007, in an attempt to clarify the sources of these pollutants. A high correlation between PCDDs/PCDFs dispersion and BP plumes emerged, indicating waste combustion as a primary source for these compounds. Correlations of PAH dispersion with plumes were more difficult to establish because of several PAHs sources in the base, such as vehicle and aircraft emissions. Rohrbeck and colleagues performed a retrospective cohort study on 200 JBB (Iraq) and Bagram Airfield (BA, Afghanistan) deployed US soldiers compared to 200 never deployed soldiers. The authors calculated relative risks (RRs) for four health encounters: obstructive sleep apnea syndrome, cardiovascular system alterations, respiratory and chest symptoms, and respiratory diseases. RRs in the deployed cohort were lower than in the non-deployed one, indicating a protective effect of deployment. RRs for respiratory diseases and for cardiovascular diseases increased when the deployed cohort was analyzed specifically for base localization. BA cohort showed an adjusted RR = 1.072 for cardiovascular diseases, while the JBB cohort showed an adjusted RR = 1.259 for respiratory diseases compared to the non-deployed cohort [77]. However, these results were not statistically significant, failing to demonstrate a role of BPs in the insurgence of specific pathologies after deployment. On the contrary, a retrospective cohort study by Abraham et al. [78] identified increased incidence rate ratios (IRRs) for respiratory symptoms and asthma in deployed soldiers compared to US-stationed soldiers (respiratory symptoms IRR = 1.25, CI: 1.20–1.30; asthma IRR = 1.54, CI: 1.33–1.78), but did not show any statistical difference between burn pit-exposed and non-exposed soldiers. These results indicate that deployment in Arabic regions, independently of burn pit exposures, is positively associated with post-deployment rates of specific respiratory conditions. Serum levels of PAHs, PCDDs, PCDFs, and the expression levels of several microRNAs were analyzed in 200 deployed soldiers’ sera before and after deployment at JBB and BA. A group of 200 non-deployed service members was used as control [79,80]. The analysis identified 56 differentially expressed microRNAs associated to generic PCDD/PCDF exposure, and no association with PAH serum levels. Among the 56, 5 microRNAs were differentially expressed after exposure to PCDD/PCDF in bases with the presence of BPs: let-7a-5p, let-7d-5p, miR-144-3p, miR-16-5p, and miR-32-5p. This result was consistent with the work performed by Dalgard and colleagues [81], who identified 5 microRNAs with different expression levels prior to and after deployment: hsa-miR-26a, hsa-miR-30b, hsa-miR-103, hsa-miR-126, and hsa-miR-766. These studies demonstrate that PAH, PCDD/PCDF serum levels, and microRNA expression pattern can potentially be used as biomarkers of exposure to BPs.

Two studies [82,83] have also investigated associations between BP exposures and birth defects in infants born by women and men deployed before and during pregnancy. Comparing the 178,766 infants born by deployed soldiers with infants born by non-deployed soldiers, no increase in odds of being diagnosed with birth defects was detected.

Falvo et al. analyzed Iraq and Afghanistan post-9/11 veterans to characterize pulmonary function. These veterans were exposed to several airborne pollutants and reported different respiratory conditions. Possible (but not measured) exposures included smoke from sulfur mine fire and burn pits, and regional dust storms. The pulmonary function tests performed on 143 veterans showed that 75% of the samples had normal lung volumes. Nevertheless, 30% of the veterans reported a reduction in lung diffusivity capacity. Since this condition is often associated with lung diseases, such as emphysema, the finding could be related to the presence of underlying lung disease instead of exposure to pollutants during veterans’ deployment [84]. Indeed, the prevalence of cigarette smoke-related lung diseases is a major confounding factor in studies dealing with biomonitoring of military personnel. This is because the prevalence of smokers was higher in the Turkish military than in civilians. In this regard, Tekbas et al. showed that 63.7% of the soldiers investigated in their study were smokers: 9.9% were occasional smokers and 53.8% were heavy smokers. Smoking frequency was higher in military personnel characterized by high education levels, high incomes, and who were sons of smoking fathers [85].

Similarly, data collected using a self-administered questionnaire indicated that the prevalence of smokers in Italian military personnel was 54.4%, while being 36.4% in civilians having the same age [86].

However, in the study on veterans deployed to Iraq and Afghanistan, DeBeer et al. showed that chronic multi-symptom illness was not related to smoke exposure [58] (Table 7).

## 7. Fuel

### 7.1. Jet Fuel

Exposure to JP-8 jet fuel, the standard aviation military fuel, can represent a specific risk factor for military personnel. Indeed, JP-8 is composed of very volatile hydrocarbon fractions. Exposure occurs primarily through inhalation of fuel vapors or cutaneous contact with liquid fuel.

A study was performed by Lemasters and colleagues [87] on 50 men working on aircraft equipment operation and maintenance at a military installation (newly hired or transferred volunteer civilian and active-duty military personnel). They measured air and expired breath concentrations of jet fuel and compounds usually found in gasoline, paints, and solvents (jet fuel (JP-4), 1,1,1-trichloroethane, methyl ethyl ketone, xylenes, toluene, and methylene chloride). Furthermore, they evaluated sister-chromatid exchanges (SCE) and micronuclei (MN) at 0, 15, and 30 days of exposure. Individuals working in the flight line (*n* = 23) and in jet fueling (*n* = 15) did not show increased SCEs or MNs, probably because their work was mainly performed outdoors. On the contrary, a significant increase in SCEs was detected in paint shop (*n* = 6) and sheet metal (*n* = 6) workers after 15 and 30 days of exposure.

The Occupational JP-8 Exposure Neuroepidemiology Study (OJENES) [88] was performed to investigate associations between JP-8 fuel-exposure and central nervous system functioning/neuro-behavioral tasks. Seventy-four USAF (US Air Force) personnel were involved in the study and divided in two groups: high and low exposed, according to their tasks. Air samples, exhaled breath, urine, and blood samples were collected, and neurological functioning tests were performed. No statistically significant differences were detected among low and high exposed groups. More papers came from the OJENES study, giving more information about specific aspects of JP-8 fuel-exposure [89,90,91].

Merchant-Borna and colleagues [90] characterized inhalation exposure on 73 USAF personnel exposed to JP-8 fuel in three different military bases. The individuals were a priori separated into two groups (high and low exposures). Personal air samples were collected over four consecutive workdays. Naphthalene and THC (total hydrocarbons) exposure varied among job classes, with higher levels for jet fueling and maintenance personnel.

A pilot study by Smith et al. [89] investigated possible roles of 1-naphthol (1N) and 2-naphthol (2N) as jet fuel exposure biomarkers in a small sample size (*n* = 24) of active-duty USAF personnel. The individuals involved in the study were a priori subdivided into three groups (high, moderate, low exposed), urinary samples were collected before and after daily shift, and personal air samples were collected through the active sampling method. Air samples were analyzed for naphthalene, while urine samples were analyzed for 1N and 2N levels. Results indicate that a priori group assignation and naphthalene levels were good predictors for 1N and 2N, suggesting that these compounds were good biomarkers for exposure to JP-8 jet fuel. In a successive study by Rodrigues and colleagues [91], the roles of 1N and 2N were confirmed on 73 USAF personnel from three different bases. Furthermore, an association between THC and post-shift urinary 2-hydroxy-fluorene concentrations was demonstrated. This study also investigated how glutathione S-transferase (GST) polymorphisms alter urinary naphthol’s, demonstrating that aerial naphthalene had a bigger effect on individuals presenting the *GSTM1*-present genotype compared to those with the *GSTM1*-null genotype.

In Maule et al. [92], VOC levels were evaluated in blood samples from 69 USAF personnel exposed to JP-8 jet fuel. They found that VOC concentrations were higher in exposed individuals and that air THC concentration was significantly associated with VOC levels in blood.

Heaton et al. [93] evaluated neurocognitive performances of 74 USAF soldiers associated with JP-8 jet fuel exposures. No significant neurocognitive decrements were detected as associated to JP-8-exposure, confirming the observations already published by Proctor et al. [88]. Fifty-seven participants were also categorized based on their noise exposure. Jet fuel-exposure was significantly associated with hearing thresholds at 4 and 8 kHz, average hearing thresholds across frequencies in the better ear. The results suggest that jet fuel-exposure, when combined with noise exposure, had an adverse effect on audibility in humans [94] (Table 8).

### 7.2. Submarine Fuel

Submariners are another category of military personnel exposed to fuel, mainly in past years, when advanced technologies were not yet adopted. They were exposed to diesel exhaust that could be responsible for acute and chronic health effects. Indeed, exhaust products, such as VOCs, benzene, nitrogen and sulfur oxides, carbon monoxide, and dioxide, among others, were inhaled, reaching the lungs, where they contributed to cardiopulmonary injury and carcinogenic process. An observational study was performed by Duplessis et al. on 38 American submariners exposed for 9 h to diesel exhaust to evaluate the development of reactive airways dysfunction syndrome (RADS) after corticosteroid treatment. This pathological condition occurs as a result of inhalation of several irritants, including diesel exhaust, and it is characterized by different symptoms (pleuritic chest pain, coughing, wheezing, headache, dyspnea, and burning sensation in eyes, nose, and throat) appearing within 24 h from exposure. The authors monitored submariners for 6 months without finding any case of RADS [95].

Another study investigated possible links between 13-year exposure to diesel exhaust of the UK Royal Navy submariners and development of chronic myeloid leukemia. Atmosphere patrol reports, derived from the submarines where the submariner was employed, were compared to epidemiological studies performed on industry workers exposed to benzene with the aim of determining the eventual submariner’s increased risk. Nevertheless, the results showed that the increased risk of developing chronic myeloid leukemia because of benzene exposure in submarines was very low [96]. However, it should be noted that only 1 submariner was examined in this study, that should be classified as a case-report only (Table 8).

## 8. Firing Ranges

Practicing in firing ranges (FRs) could result in heavy metal exposure, like lead (Pb), with inhalation as the principal route of exposure. Few studies investigated Pb contamination by FRs in military personnel.

A cross-sectional study was performed by Greenberg et al. [97] on 175 trainee soldiers of the Israel Defense Forces (IDF). Blood lead levels (BLL) were analyzed before and after the basic and the advanced training periods, together with airborne lead levels measured by personal samplers. Airborne Pb exposures were higher during the night shifts since lead concentrations are inversely related to air humidity and temperature.

At the beginning of the training, no soldiers (except one) showed detectable BLL, while lead was detected in 21% of the participants. After the basic training, the likelihood of an exposure to levels higher than 25 ug/m^3^ was 99% for 5% of the practitioners and 95% for 25% of the instructors. After advanced training, the likelihood of an exposure to levels higher than 25 ug/m^3^ was 99% for the practitioners and 90% for 10% of the instructors.

At the beginning of the training, no soldiers (except one) showed detectable BLL, while lead was detected in 21% of the participants. During basic training, 25% of instructors and 5% of trainees were exposed above the airborne lead dose of 25 μg/m^3^. During advanced training, 10% of instructors and 10% of trainees were exposed above the airborne lead dose of 25 μg/m^3^.

Accordingly, BLL was significantly higher among instructors as compared to the trainees in basic training, while this difference did not occur in advanced training.

During the same year, another study on exposure to lead in South Korean indoor FRs was published [98]. Analyses were performed on 120 subjects working or practicing at indoor firing ranges of the Republic of Korea Air Force and Navy. An overall mean of BLL = 11.3 ± 9.4 ug/dl was assessed with differences due to specific job assignment: professional shooters showed BLL levels of 14.0 ± 8.3 ug/dl, range managers 13.8 ± 11.1 ug/dl, and range supervisors 6.4 ± 3.1 ug/dl. Environmental levels of Pb were not measured in this study.

A cross-sectional study on 188 firearms instructors of the Italian State Police in indoor and outdoor firing ranges [99] highlighted an increased risk of having BLL > 100 ug/L for indoor instructors. Chronic, excessive exposure, and accumulation of neurotoxic agents such as heavy metals (lead, mercury, cadmium), and food additives such as monosodium glutamate and aspartame caused neurotoxicity and brain damage. Recent studies support a strong relationship between military-related exposure to specific neurotoxins and development of serious medical conditions and higher rates of suicide among service members [100] (Table 9).

## 9. Sunlight Exposure

Ramani and Bennet published a paper [101] in which they showed a significantly higher number among 370 Second World War veterans involved in the study were diagnosed for skin cancer after deployment in the Pacific theater rather than in the European one. These findings were consistent with higher exposure to high-intensity sunlight during the Pacific War. Indeed, sunscreen creams were invented in 1944 by Benjamin Green, airman and pharmacist, to prevent the hazards of sun overexposure in soldiers in the Pacific tropics during World War II [102].

Page and colleagues [103] performed a longitudinal follow-up study on 5524 prisoners of war and 3713 non-prisoner veterans during the Second World War. The study demonstrated that Pacific War prisoners were at higher risk for melanoma and colon cancer (OR: 3.35, 95% CI = 0.39–28.76), while the risk of melanoma in veterans from the Pacific and Europe theaters was similar (OR: 1.04, 95% CI = 0.09–11.94). Additionally, European prisoners showed higher melanoma risk than non-prisoner veterans (OR: 2.76, 95% CI = 0.31–24.81). All these results are not statistically significant, but still indicate that prisoner status was associated with higher sunlight exposure and increased melanoma risk.

In 1990, a study on active-duty US Navy personnel was published assessing an association between occupational sunlight exposure and melanoma [104]. The study was performed in the 1974–1984 period analyzing data according to three occupational groups: indoor, outdoor, and indoor and outdoor. The group with higher melanoma incidence rates compared with the general population was the indoor group (10.6 per 100,000, *p* = 0.06), while the intermediate group had the lowest rate (7.0 per 100,000, *p* = 0.06). Among the Navy personnel, two specific occupational groups were found to have high melanoma rates: aircrew survival equipment man and engineman (standardized incidence ratio (SIR): 6.8 and 2.8, respectively).

In 1998, Whiteman and Green performed a case-control study on 150 Queensland male residents with military service history, aged 50 or over, diagnosed with melanoma between 1993 and 1994. The risk of melanoma for those who served in the tropical regions for more than 3 years was 0.9 (0.3–2.7), failing in providing evidence of association between outcome insurgence and exposition to intense sunlight. Nevertheless, this could be due to the small sample size analyzed and to the very high risk of melanoma in Queensland’s general population. In these studies, the confounding role of photo-type was not evaluated. Indeed, military personnel can be composed of mixed photo-types also including high black well-protected photo-types; conversely, the Queensland general population is mainly composed of high-risk poorly protected white photo-types.

A cross-sectional study was performed by Henning and Firoz [105] on 2696 US servicemen deployed in Operation Iraqi Freedom. They detected 205 cases of skin cancer (8% prevalence), including basal and squamous cell carcinomas, mycosis fungoides, and melanoma. According to Globocan 2008, melanoma incidence rate per 100,000 inhabitants was 14.01–15.75 in the United States.

Two studies were performed on US Armed Forces to estimate incidence rates of non-melanoma skin cancer (NMSC) [106] and malignant melanoma [107]. Incidence rate for NMSC was 64.6 cases per 100,000 person-years based on the surveillance period 2005–2014 data, with a decrease between 2007 and 2013 from 68.3 to 60.4 cases per 100,000 person-years. Higher rates of NMSC were observed in white non-Hispanic subjects, males, higher ranks, and officers, and in USAF members compared to Marines. Brundage and colleagues reported an unadjusted incidence rate for malignant melanoma of 1.08 cases per 10,000 person-years during the surveillance period 2001–2015, with higher rates in pilots and crews (2.45 per 10,000 person-years), intermediate in healthcare providers group (1.33 per 10,000 person-years), and lower in infantry, special operations, and engineers (0.77 per 10,000 person-years). Furthermore, rates of malignant melanoma had a significant increase in relation to the duration of active service, with latency being shorter in pilots and crews than other occupational groups [107].

Melanoma risk for active military personnel in comparison to the general population in the USA was calculated by Lea and colleagues [108] on data from the 2000–2007 period. Melanoma risk was higher in military personnel with an incident rate ratio (IRR) of 1.62 (95% CI = 1.40–1.86). As in the articles reported before, white personnel were the most UV-sensitive ethnic group, with 98% melanoma diagnoses. White soldiers have higher incidence rates than white people in the general population (data standardized for age: 36.89 vs. 23.05 per 100,000 person-years, respectively). Both male and female soldiers had higher rates than the general US population (males: 25.32 and 16.53 per 100,000 person-years; females: 30.00 and 17.55 per 100,000 person-years). Among the different military branches, USAF personnel were confirmed to be the most subjected to melanoma, with the highest incidence rate of 17.80, while Army had the lowest, with 9.53.

The results of Lea et al. [108] are in contrast with those published by Zhou and colleagues [109] referring to US soldiers in the 1990–2004 period. This paper reported higher incident rate ratios in the general white population than in white soldiers. Overall IRR for white male soldiers was 0.75 (95% CI = 0.71–0.80) and 0.56 for white female soldiers (95% CI = 0.48–0.64). The results changed when age was considered in the analysis. In fact, IRRs rose monotonically from 0.38 in 20–24-year-old soldiers to 4.56 in 55–59-year-old personnel. IRRs were <1.0 through 40–44 years, then it turned to >1.0. The authors highlighted that melanoma incidences between 1990–1994 and 2000–2004 rose significantly both in the military and in the general population, with a faster increase in the former (36% vs. 7%). The increase was more important in young men (40% military vs. 7% general population) than in older ones (19% military vs. 12% general population). The increment of melanoma diagnosis over time can explain the IRR differences between the two studies [108,109].

Skin cancer standardized incidence ratio (SIR) and rate ratio (RR) have been calculated for Norwegian military peacekeepers deployed to Lebanon during 1978–1998 and 1999–2011. Between 1978 and 1998, SIR for skin cancer other than melanoma was 0.58, with an evident “healthy soldier effect”. In the 1999–2011 period, an elevated, not statistically significant, SIR for melanoma was observed (SIR = 1.90, 95% CI = 0.95–3.40) [110] It should be considered that UV irradiation is higher at Northern latitude (Norway) than at Southern latitude (Lebanon) due to differences in the ozone layer thickness (National Oceanic and Atmospheric Administration, NOAA), a finding contributing to explain these negative results. Moreover, a study performed by Brown et al. on 89 World War II veterans showed that the percentage of malignant melanomas, originated in nevocytic nevi, was higher in military personnel deployed in tropical theaters compared to armed forces stationed in non-tropical areas [110]. The study revealed an increased risk for melanoma associated with service in the military. Service in tropical environments was associated with an increased incidence of both melanoma and non-melanoma skin cancer among World War II soldiers. Two studies found that increased melanoma risk was also branch-dependent, with the highest rates among the United States Air Force. Several reviewed studies implicated increased sun exposure during military service and lack of sufficient sun protection as the causes of higher rates of skin cancer among US military and veteran populations as compared with non-military population in the United States [111] (Table 10).

## 10. Electromagnetic Fields

Technological breakthroughs have been leading to a progressively increasing exposure of military personnel to high-intensity radiofrequency radiation. Singh and Kapoor performed a study on 166 active soldiers of the Indian Army who were categorized in three different groups according to their exposure to electromagnetic radiations emitted from radar: group I (*n* = 40, X-band radar frequency range 8–12 GHz), group II (*n* = 58, Ku-band radar frequency range 12.5–18 GHz), and control group (*n* = 68). Blood samples were collected from participants to evaluate catecholamine concentrations (adrenaline, nor-adrenaline, and dopamine) via immunologic assay (ELISA). Moreover, electromagnetic fields’ levels were measured at different locations (inside radar cabin, at the top front of radar vehicle, and occupational spots within the 50 m range). The results showed that adrenaline levels were decreased in exposure group II while they were unchanged in group I. Significant levels were set at *p*  <  0.05. No significant differences were found in expression levels for dopamine and nor-adrenaline in both exposure groups. The authors explained the results in group II with an adaptation process that was a typical outcome after chronic radar experience. Indeed, after an initial period characterized by high stress levels and consequent enhanced production of adrenaline, biological systems tried to find a new homeostasis with reduced adrenaline levels to restore normal functioning. Regarding group I, the authors explained that radar frequency band and years of exposure were not sufficient to induce a stress response [111]. Sobiech et al. evaluated electromagnetic fields-exposure of Polish military personnel by measuring 204 devices divided into four groups: airport radars and radio navigation system, aircraft and helicopters, surveillance and height finder radars, and communication devices. The results show that Polish soldiers worked in the occupational protection zone (inside shielded cabins or away from masts with antennas) in 57% of cases, while in 43% of cases, they were not exposed to electromagnetic fields. In 35% of cases, military personnel worked in intermediate and hazardous zones [112].

Dabouis et al. performed a retrospective cohort study on 39,850 French Navy military personnel to evaluate mortality rates and specific causes of death in two groups differing in their exposure levels to radar. Occupational exposure levels derived from data collected on a French Navy ship that was considered as representative of the ships used by studied military personnel. For all causes of death, the results show that 885 deaths in the radar group and 299 in the control group occurred (RR = 1.00 (95% CI: 0.88–1.14)). RRs were 0.92 (95% CI: 0.69–1.24) for neoplasms. For the duration of follow-up, the results show that exposure to radar electromagnetic field was not responsible either for increased overall mortality or for cancer mortality [113].

Exposure to electromagnetic fields has been associated with male infertility and decreased offspring sex ratio of males to females. Baste et al., studied 10,495 military personnel employed in the Norwegian Navy and found that infertility was higher among Navy personnel who worked close to radar, high-frequency aerials, and communication equipment. Indeed, the OR among soldiers who worked closer than 10 m from high-frequency aerials compared with a control group was 1.86. Similarly, ORs for infertility of soldiers exposed to “high”, “some”, and “low” degrees were 1.93 (95% CI: 1.55–2.40), 1.52 (95% CI: 1.25–1.84), and 1.39 (95% CI: 1.15–1.68), respectively. Thus, electromagnetic radiations seemed to exert a negative effect on reproductive health that could be biologically plausible due to their thermal effects. Indeed, it is widely accepted that high temperatures were responsible for sperm quality reduction, mainly affecting its motility [114]. In addition to thermal effects, Foster et al. indicated the presence of non-thermal effects from electromagnetic radiations could affect cell membrane excitation and breakdown [115]. Decreased sperm quality was also been demonstrated for the use of laptop computers on the lap, as reported by Mortazavi et al. Indeed, besides heating men’s scrotum, laptops generate electromagnetic fields that was responsible for male infertility [116]. In addition, Wi-Fi radiofrequency emitted from modems connected to laptop computers could be responsible for reduced semen quality. Kamali et al. reported that Wi-Fi was able to reduce in vitro sperm motility and velocity without affecting sperm morphology. Mean percentages of sperm motility were not significantly different in unexposed groups (*p* = 0.22 and 0.54, respectively). In exposed semen for 50 min to 3G + Wi-Fi modem, it was significantly lower in the exposed group (*p* = 0.046) [117]. However, under these circumstances, laptops were located almost in direct touch with gonads.

Baste et al. reported a decreased offspring sex ratio of boys to girls among fathers exposed to higher degrees of electromagnetic fields [(47.6% males; OR = 0.84, 95% CI: 0.74–0.94) for soldiers who worked closer than 10 m from high-frequency aerials at very high degree of exposure level]. The authors explained this modification of sex ratio with changes in hormones. Indeed, a lower male ratio between testosterone and gonadotropin could be responsible for lower offspring sex ratio [118]. Overall, the issue regarding offspring sex ratio is controversial. Indeed, several studies performed on workers other than soldiers who were exposed to electromagnetic fields reported a decrease in male birth. Nevertheless, these studies were characterized by small sample sizes [119,120,121]. Most of the experimental studies involving mice and rats exposed to electromagnetic fields showed no significant changes in offspring sex ratio after parental exposure. The offspring sex ratios at birth in exposed and unexposed groups were 0.555 and 0.509, respectively. Statistical analyses show that there was no significant difference between the study groups for male proportion at birth (X^2^ = 0.68; *p* = 0.409) [122,123,124]. Conversely, Li et al. reported that paternal electromagnetic pulse exposure was able to significantly increase offspring sex ratio in BALB/c mice. This was clear in mice groups exposed to electromagnetic pulse at 0, 21, and 28 days after exposure compared to non-exposed sham groups. The exposure to electromagnetic fields of D0 (χ2 = 4.182, *p* = 0.0409), D21 (χ2 = 4.531, *p* = 0.033), and D28 (χ2 = 4.113, *p* = 0.0426) increased significantly compared with sham-exposed groups [125] (Table 11).

## 11. Ionizing Radiations

Since 1991, Uranium (U) has been used for Depleted Uranium (DU) military ammunitions in different war theaters. Gulf War I US veterans exposed to DU after friendly fire incidents were included in surveillance programs to investigate long-term health and genotoxic effects, as described in several studies [126,127,128,129,130]. Seventy-four Gulf War I veterans have been visited every two years since 1993 to identify health changes in response to embedded metal fragments [126]. Urine Uranium (uU) concentrations increased to be up to 1000-fold higher than in the general population. Nevertheless, the study could not detect any significant health effects due to DU-exposure. Despite high uU concentrations, retinol binding protein (RBP), used as a biomarker for renal tubule function, did not show significant differences between low and high DU-exposed individuals. Exposition class was determined on the basis of uU concentrations. Despite an absence of health effects, hypoxanthine-guanine phosphoribosyl transferases (HPRT) locus in peripheral blood indicated a weak DU-related genotoxic effect. A subgroup of 35 individuals from the same cohort of McDiarmid was analyzed for four biomarkers [126,127]. Micronuclei, chromosome aberrations, HPRT mutant frequencies, and phosphatidylinositol glycan class-A (PIGA) did not show any statistically significant differences among low-exposed and high-exposed soldiers. HPRT mutant frequencies were evaluated in 70 veterans of the cohort every 2 years (2001–2009) and no increase in mutation frequencies was detected even after 20 years since exposure (*p* = 0.61) [128]. High uU concentrations persisted even after 20 years, indicating that U is still mobilized from embedded fragments, but no health effects were observed [129]. After 25 years since DU-exposure, 36 veterans belonging to the cohort have been monitored for uU concentrations and health outcomes [130]. Isotopic uranium concentrations were elevated because of embedded fragments, while veterans who were exposed by inhalation had lower uU concentrations. Additionally, no clinical outcomes for specific target tissues (kidney and lungs) were observed, confirming the results of previous studies (*p* > 0.05). Dermatologic findings were also investigated in [131], dividing 35 veterans into two exposure groups according to uU concentrations. More than 90% of the 23 highly exposed individuals retained embedded fragments, with a significantly higher presence of fragment-related scarring. Retained fragments were documented in 91% of the high-exposure group vs. that in 13% of the low-exposure group (*p* < 0.001), and fragment-related scarring was significantly increased in the high-exposure group (*p* = 0.002). Additionally, chronic dermatitis showed a tendency to increase in the high-exposure group, but the difference was not statistically significant due to the small sample size.

Uranium concentrations were evaluated in urines and hair of 103 and 19 active and retired Canadian soldiers respectively, concerned about the effects of Uranium-exposure [132]. Thus, the study was not performed on a random sample. Urine samples were collected before deployment and after 2 months from the arrival in Kosovo or the Persian Gulf. The results, compared with data from previous studies in civilians, show that U levels were within the limits determined for non-occupationally exposed personnel.

DU exposure was investigated also in European soldiers deployed in the 1991 Gulf War, the Balkans campaign, and in the Iraq Invasion of 2003. The overall incidence of cancer was slightly higher than expected: 34 cancers were observed and 28.1 were expected based on national cancer rates, and the standardized incidence ratio = 1.2 (95% CI: 0.9 to 1.7). Among military men, there were 8 cases of testicular cancer vs. 4.6 expected [133,134,135].

A retrospective cohort study on 14,012 Danish Balkans veterans (13,552 men and 460 women) did not reveal any statistically significant variations in cancer incidence in the period 1992–2001. They found 96 cases of cancer, 84 among men (standardized incidence ratio 0.9) and 12 among women (standardized incidence ratio 1.7). Only four male bone cancers (standardized incidence ratio 6.0), with three during the first year of follow-up, exceeded expectations [135]. These data were in accordance with those obtained by Gustavsson and colleagues [133] on Swedish military personnel deployed in the same warfare theater. Again, no significant increased risk of cancer was detected compared to the civil population in the period 1989–1999. Another study on 18,175 Dutch Balkan veterans [136] show that cancer rates compared to the general population were lower in Balkan-deployed personnel (standardized incidence rate ratio 0.85 (0.73, 0.99)). The standardized incidence rate ratio for leukemia was 0.63 (0.20, 1.46), and this study showed no increased risk of cancer in deployed soldiers compared to non-deployed and civilians. Additionally, a non-significant decrease in cancer risk was observed in deployed personnel, but this effect could be explained by the “healthy warrior effect”. However, none of these studies directly assessed exposure and the latency period was too short to obtain any conclusive results. The study performed by Capocaccia et al. [137] retrospectively evaluated mortality rates for the Italian armed forces deployed in the Balkans (71,144) compared to the Italian general population, as well as Italian military personnel never operating abroad (114,269 control group). The strength of this study is mainly represented by the large sample size and the completeness of the survey; furthermore, this study has the longest follow-up period as compared to the other studies performed in European Balkan veterans. Indeed, the authors performed a detailed analysis using both the national archive of the cause of death digitized since 1999 and hardcopies for the 1995–1998 time-span. Moreover, the wide control population of military personnel (carabineers never operating abroad) was able to compensate for the healthy warrior effect by providing a comparable reference population for soldiers deployed in the Balkans. The Balkan cohort experienced mortality rates lower than both the general population (SMR = 0.56; 95% CI 0.51–0.62) and the control group (SMR = 0.88; 95% CI 0.79–0.97). Cancer mortality in the deployed cohort group was half of that of the general population mortality rates (SMR = 0.50; 95% CI 0.40–0.62) and slightly lower if compared with the control group cancer mortality rates (SMR = 0.95; 95% CI 0.77–1.18). The results reported that Balkan veterans did not show an increase neither in general mortality nor in cancer mortality [137]. Nevertheless, the main limitations of these studies are the endpoint used (mortality rate is less supporting and conclusive than incidence) and the short follow-up period (2008) compared to the typical long latency of tumors.

Bland et al. [134] analyzed a cohort of approximately 17,500 Royal Navy, British Army, and RAF personnel: 40% of the participants to this study were deployed in the 2003 Iraq War, and 341 individuals were divided a priori in four groups according to their missions, and urines were collected and analyzed. Mean ratios by group varied from 138.0 (95% CI 137.3 to 138.7) for clean-up personnel to 138.2 (95% CI 138.0 to 138.5) for combat personnel and were close to the ratio of 137.9 for natural uranium. Uranium concentration and ^238^U/^235^U ratio did not show any statistically significant difference among groups and from natural uranium levels, even in those soldiers directly exposed to DU.

A biomonitoring project involving Italian soldiers deployed in Iraq in 2004–2005 is the SIGNUM (Study of the Genotoxic Impact in Military Units). One of the main characteristics of this study is the choice of biomarkers useful to evaluate exposure to depleted uranium and other pollutants. Indeed, researchers used biomarkers of exposure and early effects in 981 soldiers, evaluating both internal doses of xenon elements and molecular doses of genotoxic compounds. Urine and blood samples were analyzed in order to evaluate the presence of DNA adducts, micronuclei frequency, and three genetic polymorphisms (GSTM1, XRCC1, and OGG1). These biomarkers are particularly informative compared to environmental monitoring as they considered interactions between the host and the environment. The study concluded that no exposition to depleted uranium occurred in Iraq during the deployment [3]. The effects of ionizing radiation were observed in soldiers who served in radar units on weapon systems that were emitting high-frequency radiation.  Multi-site de novo mutations might be suited in principle for the assessment of DNA damage from ionizing radiation in humans [138] (Table 12).

## 12. Discussion

Military personnel are exposed to different environmental pollutants that represent risk factors for the onset of different diseases. During the last 10 years, several studies have been performed to evaluate soldiers’ exposure using biomarker analyses. The evaluation of these biomarkers is part of biomonitoring programs that are composed by both biological and environmental monitoring. Indeed, a complete study should evaluate, all together, epidemiological data, environmental exposures both in the environment and as personal exposure, and biomarkers either as related to early biological response and early biological effect [139].

Nevertheless, most of the military exposure studies are incomplete if we consider biomarker analyses. Indeed, these studies focused only on biological or environmental monitoring, rarely evaluating both of them together. As an example, the studies performed on veterans exposed to sulfur mustard did not report any information on quantity of toxic agent exposure because, at the time of exposure, environmental sulfur mustard concentration was not assessed. The same situation occurs for ionizing radiations. Indeed, most of the studies considered only biological monitoring, evaluating urine, blood, and hair concentration of Uranium, while environmental monitoring was not performed.

Moreover, the choice of biomarkers is not always the most appropriate. Indeed, only few studies reported in the literature have been based on molecular epidemiology, and data obtained are limited to notice the presence of pollutants and their quantity in hosts compared to control subjects. Thus, interactions between pollutants and hosts have not been considered in most of the studies, leading to failure to assess cause–effect relationships between exposure to pollutants and disease onset. In this regard, only studies performed by Bolognesi et al., Lemaster et al., and Poirier et al. [3,69,87] considered DNA damage and micronuclei frequency as biomarkers of internal dose. In addition, several study designs did not include control groups of unexposed subjects that would be useful to make a comparison with the biomarkers in the exposed population. As an example, most of the studies performed on military personnel exposed to sulfur mustard did not have a control group, as well as studies regarding oil combustion, jet fuel, and ionizing radiations exposure.

Another critical element denoting incompleteness of some studies is represented by sample size. Considering Sarin and Cyclosarin, most of the studies had small sample sizes (13–140 exposed subjects) compared to the total number of soldiers potentially exposed, thus underestimating the actual number of exposed military personnel. The same applies to studies performed on military personnel exposed to jet fuel. As an example, the pilot study by Smith et al. [74] investigated jet fuel biomarkers in only 24 subjects belonging to active-duty US Air Force personnel.

A further limitation of many studies is the lack of health measurements. As an example, studies performed on military personnel exposed to oil fumes did not report information regarding health outcomes although these combustion products are risk factors for lung cancer. The same applies to studies regarding soldiers exposed to herbicides during the Vietnam War. Indeed, these studies failed to assess a cause–effect relationship between herbicides exposure and disease onset because they did not use direct measures of exposure and effect.

The SIGNUM (Study of the Genotoxic Impact in Military Units) project differs from the other biomonitoring programs reported in the literature primarily for the use of extremely sensitive biomarkers. In this regard, exposures to depleted Uranium and other pollutants were evaluated through the analysis of internal doses of elements or their metabolites and molecular doses of genotoxic compounds. These biomarkers are highly informative compared to environmental biomarkers, as they consider interactions between the host and the environment. Another distinctive feature of the SIGNUM project is the choice of a large sample size. Indeed, SIGNUM analyzed a cohort composed of 981 Italian military units who were deployed in Iraq in 2004–2005. The SIGNUM project was characterized by a large sample size. Indeed, the only other biomonitoring study performed in military personnel using ^32^P post-labeling analyzed only 22 soldiers [69]. The SIGNUM study was designed on a follow-up basis analyzing the same endpoints in the same subjects either before (T0) or at the end of the deployment. This approach allows accurate control of the multiple confounding factors affecting comparison of epidemiological and biomarkers in different groups. These procedures were combined with the use of sensitive biomarkers, such as ^32^P post-labeling which was able to detect up to 1 single nucleotide modification every 10 cells. Indeed, environmental pollution does not induce generalized health effects on the whole population but mainly on specific subgroups of fragile subjects referred to as ‘susceptible’. In SIGNUM, these susceptible individuals have been identified by analyzing a combination of SNPs genotypes. The combination of GSTM1-null and OGG1-slow SNPs identified susceptible subjects undergoing a slight increase of oxidative biomarkers between T0 and T1. However, this combination is extremely rare, occurring in less than 3% of the population examined. It should be noted that oxidative biomarkers, when occurring alone, do not reflect occurrence of biological damage but indicate the adaptive response of the organism to an environmental challenge. However, the SIGNUM follow-up has been extended specifically to further investigate this aspect.

## 13. Conclusions

Military personnel are exposed to a wide variety of specific occupational and environmental pollutants that can increase risk for diseases. Accordingly, biomonitoring of soldiers is a unique opportunity to study health effects of pollutants in humans and to address preventive intervention in the whole population.

Therefore, much better experimental design and selection of biomarkers are needed to address health hazards of military personnel.

Overall, the reported data provide strong justification that epidemiologic studies analyzing health indicators and molecular epidemiology studies analyzing molecular biomarkers should be performed together to investigate occupational and environmental health risks of military personnel. Indeed, only the integration between epidemiology and biomonitoring can produce results applicable for preventive medicine purposes. Biomonitoring can provide, on a short-term basis, evidence (or lack of) of biological effects, thus triggering the activation of preventive programs to decrease risks. Epidemiology can provide, on a long-term basis, evidence (or lack of) of effects on incidence of non-communicable diseases (cancer, cardiovascular, COPD). Due to existence of many confounding factors, only the integration between these two approaches can provide clear-cut responses to answers dealing with the possible health risks of military personnel. Establishing a temporal relationship is often difficult, especially with health outcomes that have long induction periods, such as cancer. Thus, military personnel may have been exposed to a variety of agents, at varying doses and lengths of time. The literature on the agents, however, is quite limited with regard to combinations of biological and chemical agents and their interactions.

Military personnel are a particular category of subjects differing from the general population for various reasons, including the healthy warrior effect as well as accurate health assessment. To implement prevention, it is essential to design monitoring programs based on the use of sensitive and informative biomarkers which are able to evaluate not only exposure but especially interactions between pollutants and the host.

The use of integrated studies analyzing both epidemiological variables and biomarkers is the most suitable tool to address these problems.

## Figures and Tables

**Figure 1 ijerph-18-05395-f001:**
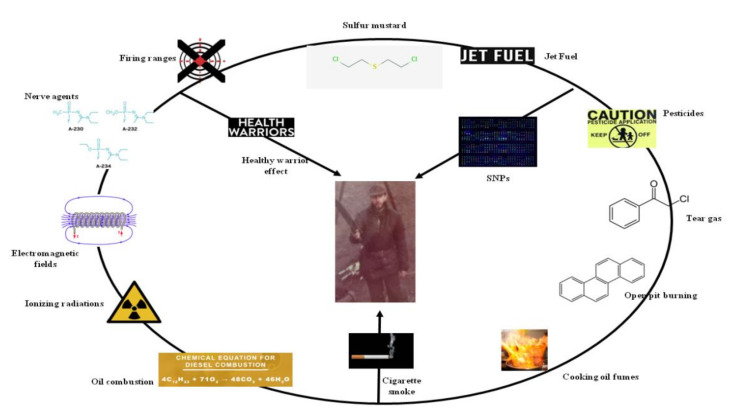
Possible exposures of military personnel to chemical and physical occupational hazards.

**Figure 2 ijerph-18-05395-f002:**
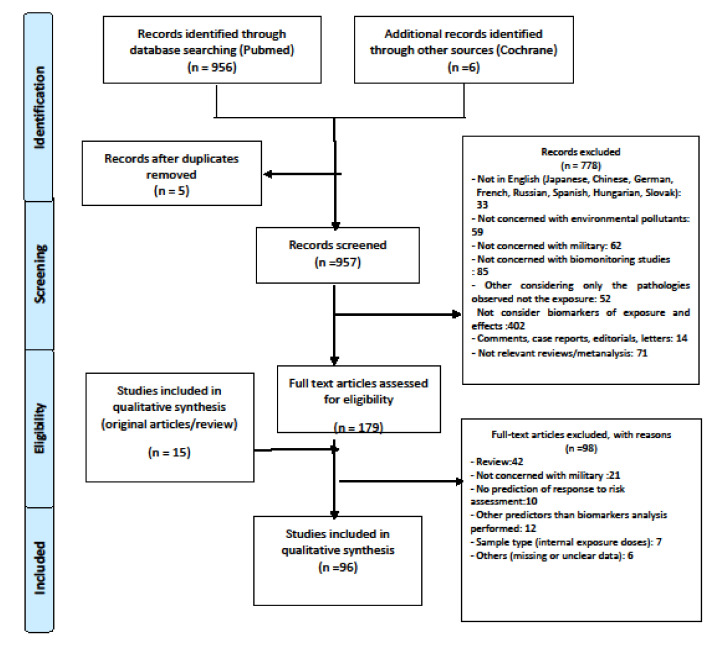
Flow chart reporting the procedure used to perform the systematic review.

**Table 1 ijerph-18-05395-t001:** Overview of the main studies dealing with the sulfur mustard exposures of military personnel.

Pollutant	Sample Size	Time after Exposure	Solider Nationality	Exposure Site (Operation Theater)	Environmental Monitoring (Exposure)	Biomarkers (Early Biological Effect)	Biomarker Description	Early Clinical Effect	Late Clinical Effect	References
Sulfur Mustard	236	2–28 months	Iran	Iran/Iraq border	-	-	-	Respiratory tract, central nervous system, skin, eyes.	-	[17]
	75	16–20 years	Iran	Khorasan (Iran)	-	Immunity	WBC, RBC, hematocrit, IgM, C3, monocytes, and CD3+ lymphocytes CD16+.	-	Complications in the lungs, peripheral nerves, skin, and eyes.	[19]
	134	17–22 years	Iran	Fars (Iran)	-	-	-	-	Complications in lung, skin, and eyes.	[20]
	43	12–17 years	Iran	Khorasan (Iran)	-	-	-	-	Complications in lungs, peripheral nerves, skin, eyes, head, and neck.	[21]
	12	6–8 days	Iran	Majnoun Island and Persian Gulf	-	-	-	Eye, throat, lung, and gastro-intestinal symptoms	-	[12]
	372	20 years	Iran	Sardasht (Azerbaijan)	-	Immunity	Chemokines, citokines, and inflammatory markers	-	Respiratory function	[15,16]
	40	16–20 years	Iran	Khorasan (Iran)	-	-	-	-	COPD, bronchiectasis, asthma, large airway narrowing, pulmonary fibrosis, simple chronic bronchitis.	[23]
	43	20–25 years	Iran	Iran/Iraq border	-	-	-	-	Dysphonia, post-nasal discharge, lower larynx position, limitation of vocal cords, inflammation of larynx mucosa, COPD, asthma, large-airway narrowing, pulmonary fibrosis, simple chronic bronchitis.	[24]
	43	25 years	Iran	Iran/Iraq border	-	-	-	-	Bronchiectasis, pulmonary fibrosis, and ground-glass attenuation.	[22]
	75	16–20 years	Iran	Khorasan (Iran)	-	Immunity	WBC, RBC, Hct, monocytes, CD3+T-lymphocytes, CD16+56 positive cells, IgM and C3 levels.	-	Increased risk of infections and tumors.	[27]
	200	-	Iran	Iran	-	-	-	-	Increased prevalence of oral candidiasis.	[28]
	11	25 years	Iran	Iran	-	Gene expression	Peroxiredoxins (PRDXS) and sulfiredoxin-1 (SRXN1), oxidative stress responsive kinase-1 (OXSR1), forkhead box M1 (FOXM1), glutathione peroxidase-2 (GPX2), aldehyde oxidase 1 (AOX1), myeloperoxidase (MPO), dual oxidase 1 and 2 (DUOX1, DUOX2), thyroid peroxidase (TPO), eosinophil peroxidase (EPO), metallothionein-3 (MT3), and glutathione reductase.	-	-	[29,30,31]
	268	25 years	Iran	Iran	-	Oxidative stress, gene expression, cellular senescence	Lipid peroxidation derivative malondialdehyde (MDA), 8-oxo-dG, OGG1, and p16^INK4a^ mRNA.	-	DNA damage and immune system subjected to cellular senescence.	[33]
	80	-	Iran	Iran	-	Oxidative stress	Prostaglandin-like compound 8-isoprostane F2- alpha	-	-	[34]
	75	25 years	Iran	Khorasan (Iran)	-	DNA damage	DNA repair proteins (MRE11, NBS1, RAD51, and XPA), phosphor-H2AX.	-	Long-term health problems.	[18]
	81	-	Iran	Iran	-	microRNA expression	miR-589-3p, miR-365a-3p, miR-143-3p, miR-200a-3p, miR-663a.	-	-	[35]
	34,000	18-23 years	Iran	Iran		-	-	-	Lesions of the lungs, eyes and skin	[36]		

**Table 2 ijerph-18-05395-t002:** Overview of the main studies dealing with Sarin and Cyclosarin exposures of military personnel.

Pollutant	Sample Size	Time after Exposure	Solider Nationality	Exposure site (Operation Theater)	Environmental Monitoring (Exposure)	Biomarkers (Early Biological Effect)	Biomarker Description	Early Clinical Effect	Late Clinical Effect	References
Sarin and Cyclosarin	349,291	-	United States	Khamisiyah (Iraq)	-	-	-	-	Suffering postwar morbidity.	[37]
	1,368,150	7 years	United States	Khamisiyah (Iraq)	-	-	-	-	Cause-specific mortality.	[38]
	80	11-16 years	United States	Khamisiyah (Iraq)	-	-	-	-	Reduced total gray matter volume.	[40]
	128	14–19 years	United States	Khamisiyah (Iraq)	-	-	-	-	Reduced gray matter and white matter volumes.	[41]
	118	14–19 years	United States	Khamisiyah (Iraq)	-	-	-	-	Increased axial diffusivity throughout the brain.	[42]
	170	23–26 years	United States	Khamisiyah (Iraq)	-	-	-	-	Smaller hippocampal volumes.	[43]
	351,041	10 years	United States	Khamisiyah (Iraq)	-	-	-	-	Increased risk of brain cancer death.	[39]
	26	-	United States	Khamisiyah (Iraq)	-	-	-	-	White matter reduction and ventricles volume increase.	[44]
	140	4–5 years	United States	Khamisiyah (Iraq)	High exposure: 0.072 mg min/m^3^; Moderate exposure: from 0.035 to 0.144 mg min/m^3^ but no more than 0.072 mg min/m^3^	-	-	-	Neuropsychological task performances for psycho-motor dexterity and visuospatial abilities.	[45]

**Table 3 ijerph-18-05395-t003:** Overview of the main studies dealing with the Agent Orange exposures of military personnel.

Pollutant.	Sample Size	Time after Exposure	Solider Nationality	Exposure Site (Operation Theater)	Environmental Monitoring (Exposure)	Biomarkers (Early Biological Effect)	Biomarker Description	Early Clinical Effect	Late Clinical Effect	References
Agent Orange	-	-	United States	Vietnam	Inner aircraft concentrations ranged from 11.49 to 13.2–27.0 pg/m^3^	-	-	-	-	[48]
	-	-	United States	Vietnam	Exposure estimated using a software	-	-	-	-	[49]
	5609	-	United States	Vietnam/ Non-Vietnam	-	-	-	-	COPD mortality.	[50]
	111,726	-	Korea	Vietnam	Exposure indexes based on the proximity of the veterans’ military unit to an Agent Orange-sprayed area.	-	-	-	Diabetes mellitus, thyroid, pituitary gland, and neurologic disorders.	[51]
	180,639	30 years	Korea	Vietnam	Exposure indexes based on the proximity of the veterans’ military unit to an Agent Orange-sprayed area.	-	-	-	Cancers of the stomach, small intestine, liver, larynx, lung, bladder, and thyroid gland, chronic myeloid leukemia.	[52]

**Table 4 ijerph-18-05395-t004:** Overview of the main studies dealing with the pesticides exposures of military personnel.

Pollutant	Sample Size	Time after Exposure	Solider Nationality	Exposure Site (Operation Theater)	Environmental Monitoring (Exposure)	Biomarkers (Early Biological Effect)	Biomarker Description	Early Clinical Effect	Late Clinical Effect	References
Pesticides	187	Day 0, 14, and 28 of the wearing period, and 28 days after termination	Germany	Germany	Permethrin concentration in uniforms: 1300 mg/m^2^	Exposure biomarker in urines	Permethrin metabolites: cis-3-(2,2-dichlorovinyl)-2,2-dimethyl-(1-cyclopropane) carboxylic acid (cis-DCCA), trans-3-(2,2-dichlorovinyl)-2,2-dimethyl-(1-cyclopropane) carboxylic acid (trans-DCCA), and 3-phenoxybenzoic acid (3-PBA)	-	-	[55,56]
	549 (Study I); 195 (Study II)	Before wearing uniforms, after 14 days, after 28 days of wearing, and 28 days after cessation of wearing uniform	Germany	Two sub-cohorts in Germany and one in Afghanistan	Permethrin concentration in uniforms: 0.13 mg/cm^2^	Exposure biomarker in urines	Permethrin metabolites: cis-DCCA, trans-DCCA 3-PBA	-	-	[57]
	224	-	United States	Iraq/Afghanistan	-	-	-	-	Chronic multisymptom illness.	[58]

**Table 5 ijerph-18-05395-t005:** Overview of the main studies dealing with the tear gas-exposures of military personnel.

Pollutant	Sample Size	Time after Exposure	Solider Nationality	Exposure site (Operation Theater)	Environmental Monitoring (Exposure)	Biomarkers (Early Biological Effect)	Biomarker Description	Early Clinical Effect	Late Clinical Effect	References
Cs Gas (Tear Gas)	87	2–8–24–30 h	United States	United States	0.086–4.9 mg/m^3^	Exposure biomarker in urines	2-chlorohippuric acid	-	-	[60]
	-	-	United States	United States	2.33–3.29 mg/m^3^	-	-	-	-	[61]
	6730	5–15 min during exposure	United States	United States	0.4–53.3 mg/m^3^	Exposure biomarker	Personal air sampling	-	-	[62]
	6723	7 days before CS exposure and 7 days after exposure	United States	United States	0–2 mg/m^3^, 2–5 mg/m^3^, 5–10 mg/m^3^, >10 mg/m^3^	-	-	-	Acute respiratory illnesses.	[63]

**Table 6 ijerph-18-05395-t006:** Overview of the main studies dealing with the combustion products exposures of military personnel.

Pollutant	Sample Size	Time after Exposure	Solider Nationality	Exposure Site (Operation Theater)	Environmental Monitoring (Exposure)	Biomarkers (Early Biological Effect)	Biomarker Description	Early Clinical Effect	Late Clinical Effect	References
Oil Combustion	1599	3 months	United States	Kuwait	-	-	-	Eye and upper respiratory tract irritation, shortness of breath, cough, rashes, and fatigue	-	[64]
	125	During exposure: every fortnight for 5 months	United Kingdom	Kuwait	-	-	-	-	No lung function changes before and after deployment.	[65]
	1560	5 years	United States	Kuwait	-	-	-	-	Asthma, bronchitis, injury and major depression.	[66]
	61	Sampling: before, during, and after deployment in Kuwait	Germany	Kuwait	-	DNA adducts and gene polymorphism	PAH-DNA adducts in blood; -OH-PG (1-hydroxypyrene-glucuronide) in urines; GSTM1 and GSTT1 gene polymorphisms	-	-	[68]
	168	-	United States	Kuwait city	-	Exposure biomarkers in blood	VOC: benzene, m-/p-xylene, o-xylene, styrene, and toluene	-	-	[68]
	3440	Sampling: before, during, and after deployment in Kuwait	Germany	Kuwait	-	-	1-hydroxypyrene; VOCs; Sister chromatid exchange; DNA-PAH adducts.	-	-	[70]
Cooking Oil Fumes	98	At the beginning and at the end of the weekly shift	Taiwan	Taiwan	PAH measured in kitchens and in offices	Exposure biomarker in urines and oxidative DNA damage biomarker	1-hydroxypyreneand 8-oxodeoxyguanosine	-	Oxidative DNA damage.	[71]

**Table 7 ijerph-18-05395-t007:** Overview of the main studies dealing with the open pit burning-exposures of military personnel.

Pollutant	Sample Size	Time after Exposure	Solider Nationality	Exposure site (Operation Theater)	Environmental Monitoring (Exposure)	Biomarkers (Early Biological Effect)	Biomarker Description	Early Clinical Effect	Late Clinical Effect	References
Open Pit Burning	20,000	1.3 years (mean years)	United States	Iraq and Afghanistan	-	-	-	-	Rheumatoid arthritis and lupus.	[72]
	21,000	-	United States	Iraq	-	-	-	-	Chronic multisymptom illness.	[73]
	22,844	-	United States	Iraq	-	-	-	-	Respiratory outcomes.	[74]
	-	-	-	Iraq	PAHs, polychlorinated dibenzo-p-dioxins (PCDD), and polychlorinated dibenzofurans (PCDF)	-	-	-		[75,76]
	400	-	United States	Iraq	-	-	-	-	Respiratory and cardiovascular disease.	[77]
	179,914	-	United States	Iraq, United States, and South Korea	-	-	-	-	Respiratory symptoms and asthma.	[78]
	400	-	United States	Iraq and Afghanistan	-	microRNA expression	PCDD/PCDF: let-7a-5p, let-7d-5p, miR-144-3p, miR-16-5p, miR-32-5p	-	-	[79]
	400	-	United States	Iraq and Afghanistan	-	Exposure biomarkers in serum	PAH and PCDD/PCDF	-	-	[80]
	800	-	-	-	-	microRNA expression	hsa-miR-26a, hsa-miR-30b, hsa-miR-103, hsa-miR-126, hsa-miR-766.	-	-	[81]
	178,766 infants	-	-	-	-	-	-	-	Birth defects.	[82]

**Table 8 ijerph-18-05395-t008:** Overview of the main studies dealing with the fuel-exposures of military personnel.

Pollutant	Sample Size	Time after Exposure	Solider Nationality	Exposure Site (Operation Theater)	Environmental Monitoring (Exposure)	Biomarkers (Early Biological Effect)	Biomarker Description	Early Clinical Effect	Late Clinical Effect	References
Jet Fuel	130	Prior to and after exposure, 15 and 30 weeks during exposure.	United States	United States	Jet fuel (JP-4), 1,1,1- trichloroethane, methyl ethyl ketone, xylenes, toluene, and methylene chloride.	Biomarkers of genotoxicity	Chromatid exchanges and micronuclei frequency	-	-	[87]
	74	Post-shift on friday (Day 1) and during the following workweek (Days 2–6)	United States	United States	Personal breathing zone and work areas via active sampling methods.	Exposure biomarkers in urines, exhaled breath, and blood. Gene polymorphism	- Dermal exposure (total hydrocarbons, benzene, toluene, ethylbenzene, and xylene and naphthalene); - Exhaled breath (BTEX and naphthalene); -Urine samples (1- and 2-naphthol, 2-, 3-, and 9-hydroxyfluorene; 1-, 2-, 3-, and 4-hydro-xyphenanthrene; 1-hydroxypyrene and VOC mercapurates, including N-acetyl-S-(benzyl)-L-cysteine, N-ace-tyl-S-(phenyl)-L-cysteine. - Blood samples (benzene, ethylbenzene, m-/p-/o-xylenes, and toluene). - GSMT1 polymorphism and DNA pattern methylation	-	-	[88]
	24	During three consecutive workdays	United States	United States	Personal air samples: naphthalene	Biomarker of exposure in urines	1- and 2-naphthol	-	-	[89]
	73	During four consecutive workdays	United States	United States	Personal air samples: benzene, ethylbenzene, toluene, xylene, total hydrocarbons, and naphthalene.	-	-	-	-	[90]
	73	During four consecutive workdays	United States	United States	Personal air samples analyzed for benzene, toluene, ethylbenzene, m-/p-xylene, o-xylene, and total hydrocarbons	Biomarker of exposure in urines. Gene polymorphism	1- and 2-naphthol, 2-, 3-, and 9-hydroxyfluorene, 1-, 2-, 3-, and 4-hydroxyphenanthrene, and 1-hydroxypyrene; GSTM1 polymorphism and glutathione S-transferase theta-1 (GSTT1)	-	-	[91]
	69	Post-shift sampling	United States	United States	Personal air samples: total hydrocarbons	Biomarker of exposure in blood	Ethylbenzene, toluene, o-xylene, and m/p-xylene, and for the smoking biomarker, 2,5-dimethylfuran.	-	-	[92]
	74	Post-shift on a Friday afternoon (Day1) and continued Monday morning through Friday morning of the following workweek (Days 2–6)	United States	United States	Personal air samples: naphthalene and total hydrocarbons.	Biomarkers of exposure in urines	1-naphthol and 2-naphthol	-	-	[93]
Submarine fuel	38	-	United States	United States	-	-	-	Reactive airways dysfunction syndrome (RADS)	-	[95]
	1	-	United Kingdom	-	Median weighted levels of benzene exposure over 13-year period: 189 ug/m^3^	-	-	-	Chronic myeloid leukemia.	[96]

**Table 9 ijerph-18-05395-t009:** Overview of the main studies dealing with the firing ranges-exposures of military personnel.

Pollutant	Sample Size	Time after Exposure	Solider Nationality	Exposure Site (Operation Theater)	Environmental Monitoring (Exposure)	Biomarkers (Early Biological Effect)	Biomarker Description	Early Clinical Effect	Late Clinical Effect	References
Firing Ranges	175	During training period	Israel	Israel	Overall range of airborne lead levels in personal samples: 0.08–168.4 μg/m^3^	Biomarkers of exposure in blood	Blood lead level	-	-	[97]
	120	During working period	Korea	Korea	-	Biomarkers of exposure in blood	Blood lead level	-	-	[98]
	546	During working period	Italy	Italy	Environmental lead using personal samplers in 6 indoor and 6 outdoor firing ranges (<25 pg/m^3^)	Biomarkers of exposure in blood	Blood lead level	-	-	[99]

**Table 10 ijerph-18-05395-t010:** Overview of the main studies dealing with the sunlight exposures of military personnel.

Pollutant	Sample Size	Time after Exposure	Solider Nationality	Exposure Site (Operation Theater)	Environmental Monitoring (Exposure)	Biomarkers (Early Biological Effect)	Biomarker Description	Early Clinical Effect	Late Clinical Effect	References
Sunlight exposure (Skin Cancer)	154	-	United States	Tropical/non-tropical theaters	-	-	-	-	Melanoma	[110]
	370	-	-	Pacific theater (World War II)	-	-	-	-	Skin cancer.	[101]
	5524	-	-	Pacific theater (World War II)	-	-	-	-	Melanoma and colon cancer.	[103]
	176	During working period	United States	United States	-	-	-	-	Melanoma	[104]
	300									
	2696	-	-	Iraq	-	-	-	-	Eczematous dermatitis, benign neoplasms, and skin cancers.	[105]
	6670	-	United States	United States	-	-	-	-	Non-melanoma skin cancer.	[106]
	-	-	United States	United States	-	-	-	-	Melanoma.	[107]
	2093,157	-	United States	United States	-	-	-	-	Melanoma.	[108]
	35,157	-	Australia	Tropical locations	-	-	-	-	Melanoma.	[36]
	21,582	During exposure: from first day of service until date of death, emigration, or end of follow-up.	Norway	Lebanon	-	-	-	-	Cancer incidence and all-cause mortality	[111]

**Table 11 ijerph-18-05395-t011:** Overview of the main studies dealing with the electromagnetic fields exposures of military personnel.

Pollutant	Sample Size	Time after Exposure	Solider Nationality	Exposure Site (Operation Theater)	Environmental Monitoring (Exposure)	Biomarkers (Early Biological Effect)	Biomarker Description	Early Clinical Effect	Late Clinical Effect	References
Electromagnetic fields	166	During exposure	India	India	Electromagnetic fields monitored at different locations.	Stress response	Catecholamine levels in blood	-	-	[111]
	-	-	Poland	Poland	204 military devices monitored	-	-	-	-	[112]
	39,850	-	France	France	-	-	-	-	Overall mortality and cancer mortality	[113]
	10,495	-	Norway	Norway	-	-	-	-	Male infertility and offspring sex ratio	[118]

**Table 12 ijerph-18-05395-t012:** Overview of the main studies dealing with the ionizing radiation exposures of military personnel.

Pollutant	Sample Size	Time after Exposure	Solider Nationality	Exposure Site (Operation Theater)	Environmental Monitoring (Exposure)	Biomarkers (Early Biological Effect)	Biomarker Description	Early Clinical Effect	Late Clinical Effect	References
Ionizing Radiations	74	2 years	United States	Iraq	-	Biomarker of exposure in urines	Uranium concentration	-	-	[129]
	35	3 years	United States	Iraq		Biomarker of exposure in urines	Total and isotopic uranium concentrations. Analysis of metals.	-	-	[129]
	35	18 years	United States	Iraq	-	Biomarkers of exposure in urines. Gene mutations.	Total and isotopic uranium concentration; Peripheral blood cells: micronuclei (MN), chromosome aberrations, and mutations of hypoxanthine- guanine phosphoribosyl transferase (HPRT) and phosphatidylinositol glycan class-A (PIGA).	-	-	[126]
	36	25 years	United States	Iraq	-	Biomarkers of exposure in urines	Total and isotopic uranium concentrations. Analysis of metals.	-	-	[130]
	70	-	United States	Iraq	-	Biomarkers of exposure in urines. Gene mutations	Total and isotopic uranium concentrations. Analysis of metals. Peripheral blood cells: hypoxanthine-guanine phosphoribosyl transferase (HPRT) mutations	-	-	[128]
	35	18 years	United States	Iraq	-	Biomarker of exposure in urines and biomarkers of genotoxicity	Total uranium concentration Peripheral blood cells: Micronuclei	-	-	[127]
	35	22 years	United States	Iraq	-	Biomarker of exposure in urines.	Total uranium concentration	-	chronic dermatological findings (dermatitis)	[131]
	122	-	Canada	Canada	-	Biomarker of exposure in urines.	Total uranium concentration Hair: isotopic uranium (238:235 U ratio)	-	-	[132]
	9188	-	Sweden	Balkans	-	-	-	-	Overall incidence of cancer	[133]
	341	-	United Kingdom	Iraq	-	Biomarker of exposure in urines.	Uranium concentration and 238 U/235 U isotopic ratio	-	-	[134]
	14,012	-	Denmark	Balkans	-	-	-	-	Overall incidence of cancer	[135]
	153,530	-	Holland	Balkans	-	-	-	-	incidence of all cancer	[136]
	185,413	4–14 years	Italy	Balkans	-	-	-	-	General mortality and cancer mortality	[137]
	981	-Before the deployment (T0) -At the end of the deployment period (T1) - In the operative theater	Italy	Iraq	-	Biomarkers of exposure in urine and serum. Biomarkers of genotoxicity and DNA damage	- Urine and serum: As, Cd, Mo, Ni, Pb, U, V, W, and Zr, and Uranium isotopic ratio 238 U/235 U Peripheral blood cells: DNA adducts, micronuclei, GSTM1, XRCC1, and OGG1 polymorphisms; Oxidative DNA alterations (8-Hydroxy-2′-deoxyguanine)	-	-	[3]

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
