# Peer review of "Occupational Exposures and Environmental Health Hazards of Military Personnel"

_ijerph, 2021, doi:10.3390/ijerph18105395_

Round 1
Reviewer 1 Report
The authors have performed a review of the epidemiological studies describing exposures of military personnel. The authors concluded that only studies analyzing in the same population both epidemiological and molecular biomarkers provide evidence-based conclusion suitable to support or deny the existence of a relationship between on field exposure to pollutants and adverse health consequences in military personnel.
This conclusion is quite general and extendible to all occupational and environmental exposures and therefore, in this sense, the conclusion is not novel or disruptive. Nevertheless, the review is well performed and I have no objections as such. However, I have a few number of comments addressed to formal aspects.
The authors must do an effort in order to make the reading friendlier. I suggest, by example, to split Table 1 in smaller tables, one per each of the eleven reviewed physical or chemical agents. It is very hard to scrutiny a 20-pages long table and it is hard as well to read 25 pages of plane text.
I missed in the manuscript a description of exposures to pyridostigmine bromide used as prophylactic treatment against potential exposure to warfare agents and potentially involved in the so-called, Gulf War Illness. Why this case was excluded from the review? Could the authors include something about this issue?
I have doubts about whether the term environmental hazard is appropriate because all the situations reviewed in the manuscript are addressed to occupational exposures.
It is necessary to handle the use of abbreviations more carefully. I have found several abbreviations without definition; by example Fas-L or VOC. The term PCDD is used twice before the third citation where is indeed defined.
Figure 2 suggests that the number of studies in qualitative synthesis is of 66. However, the number of references in the reference list is 143 and the number of references included in Table 1 is around 100. There is some kind of mismatches among these figures. Please, clarify.
I do not totally agree with the sentence “Toxicity of many of these substances is not well-characterized” (lines 65-66). Maybe the epidemiology is not well-known, but the toxicity is well characterized through animal toxicity studies.
Author Response
Reviewer 1:
COMMENT 1. This conclusion is quite general and extendible to all occupational and environmental exposures and therefore, in this sense, the conclusion is not novel or disruptive. Nevertheless, the review is well performed and I have no objections as such.
ANSWER 1. The conclusion has been now more focused on the peculiarity of analysing the exposure effect relationship in military personnel.
COMMENT 2. The authors must do an effort in order to make the reading friendlier. I suggest, by example, to split Table 1 in smaller tables, one per each of the eleven reviewed physical or chemical agents. It is very hard to scrutiny a 20-pages long table and it is hard as well to read 25 pages of plane text.
ANSWER 2. Table 1 has been replaced with 12 smaller tables one per each agent as requested by the reviewer.
COMMENT 3. I missed in the manuscript a description of exposures to pyridostigmine bromide used as prophylactic treatment against potential exposure to warfare agents and potentially involved in the so-called, Gulf War Illness. Why this case was excluded from the review? Could the authors include something about this issue?
ANSWER 3. We appreciated the Reviewers’ comments, and we added a paragraph in the section 4 Sarin and Cyclosarin about the exposures to pyridostigmine bromide.
COMMENT 4. I have doubts about whether the term environmental hazard is appropriate because all the situations reviewed in the manuscript are addressed to occupational exposures.
ANSWER 4. ‘environmental hazard’ has been changed into ‘occupational hazard’.
COMMENT 5. It is necessary to handle the use of abbreviations more carefully. I have found several abbreviations without definition; by example Fas-L or VOC. The term PCDD is used twice before the third citation where is indeed defined.
ANSWER 5. The abbreviations have been defined and corrected in the text.
COMMENT 6. Figure 2 suggests that the number of studies in qualitative synthesis is of 66. However, the number of references in the reference list is 143 and the number of references included in Table 1 is around 100. There is some kind of mismatches among these figures. Please, clarify
ANSWER 6. We appreciated the Reviewers’ comments, we revised the Figure 2, it has been corrected and the number reported in the studies in qualitative synthesis is of 96, the same number of the references included in all tables.
COMMENT 7. I do not totally agree with the sentence “Toxicity of many of these substances is not well-characterized” (lines 65-66). Maybe the epidemiology is not well-known, but the toxicity is well characterized through animal toxicity studies.
ANSWER 7. The sentence has been corrected.

Reviewer 2 Report
This manuscript try to cover a lot of data more or less specific to military personel.
It is mainly based on the National Library of Medicine’s PubMed online alogue and the 143 Cochrane Database of Systematic Reviews from inception up to 1964; an update was 144 carried out on 2020. Still there are aother papers included in the discussion. Regarding for example Sarin exposure in the Tokyo subway.
I am not really convinced regardign the search strategy. I agree on the overall aim that th studies should be relevant for military personel.
Minor line 167 and 168 Figure 2. is repeated and the number n=947 near the lowest part of the figure seems misplaced.
Line 675-676 The expresssion "After the basic training, the likelihood of
an exposure to levels higher than 25ug/m3 was 99% for 5% of the practitioners and 95% 676 for 25% of the instructors." and lines 677-678 "the likelihood of an exposure to levels higher than 25ug/m3 was 99% for the practitioners and 90% for 10% of the instructors." needs more explanation to be readable.
Overall there are diffrents styles used presenting the different exposure and risk estamates sometimes Risk estimates and confidenc intervalls are given in other casees only wordning like no increased risk is used. Please treat the data in a similar way as long as possible or at least explain if you present them differently. See for example section 10 on electromagnetic fields.
Author Response
Reviewer 2:
COMMENT 1. Minor line 167 and 168 Figure 2. is repeated and the number n=947 near the lowest part of the figure seems misplaced.
ANSWER 1. The repetitions have been corrected as suggested
COMMENT 2. Line 675-676 The expression "After the basic training, the likelihood of
an exposure to levels higher than 25ug/m3 was 99% for 5% of the practitioners and 95% 676 for 25% of the instructors." and lines 677-678 "the likelihood of an exposure to levels higher than 25ug/m3 was 99% for the practitioners and 90% for 10% of the instructors." needs more explanation to be readable.
ANSWER 2. The sentence was fully rephrased.
COMMENT 3. Overall there are different styles used presenting the different exposure and risk estimates sometimes Risk estimates and confident intervals are given in other cases only wordning like no increased risk is used. Please treat the data in a similar way as long as possible or at least explain if you present them differently. See for example section 10 on electromagnetic fields.
ANSWER 3. We appreciated the Reviewers’ comments, we revised the manuscript and in particular section 10 on electromagnetic fields as suggested. The changes in the revised text are marked in yellow.

Round 2
Reviewer 1 Report
The authors have successfully addressed all the concerns that I raised in my first report. Thus, I have no further objections against the publication of the manuscript.